# Neutralization of SARS-CoV-2 by highly potent, hyperthermostable, and mutation-tolerant nanobodies

Thomas Güttler[1,†] , Metin Aksu[1,†] , Antje Dickmanns[2], Kim M. Stegmann[2] , Kathrin Gregor[1], Renate Rees[1], Waltraud Taxer[1], Oleh Rymarenko[1] , Jürgen Schünemann[1], Christian Dienemann[3], Philip Gunkel[1] , Bianka Mussil[1], Jens Krull[1], Ulrike Teichmann[4] , Uwe Groß[5], Volker C Cordes[1] , Matthias Dobbelstein[2,*] & Dirk Görlich[1,**]

## Abstract

**Monoclonal anti-SARS-CoV-2 immunoglobulins represent a treatment option for COVID-19. However, their production in mammalian cells is not scalable to meet the global demand. Single-domain (VHH) antibodies (also called nanobodies) provide an alternative suitable for microbial production. Using alpaca immune libraries against the receptor-binding domain (RBD) of the SARS-CoV-2 Spike protein, we isolated 45 infection-blocking VHH antibodies. These include nanobodies that can withstand 95°C. The most effective VHH antibody neutralizes SARS-CoV-2 at 17–50 pM concentration (0.2–0.7 µg per liter), binds the open and closed states of the Spike, and shows a tight RBD interaction in the X-ray and cryo-EM structures. The best VHH trimers neutralize even at 40 ng per liter. We constructed nanobody tandems and identified nanobody monomers that tolerate the K417N/T, E484K, N501Y, and L452R immune-escape mutations found in the Alpha, Beta, Gamma, Epsilon, Iota, and Delta/Kappa lineages. We also demonstrate neutralization of the Beta strain at low-picomolar VHH concentrations. We further discovered VHH antibodies that enforce native folding of the RBD in the *E. coli* cytosol, where its folding normally fails. Such "fold-promoting" nanobodies may allow for simplified production of vaccines and their adaptation to viral escape-mutations.**

**Keywords** Coronaviridae; COVID-19; nanobody; SARS-CoV-2; VHH antibody
**Subject Categories** Immunology; Microbiology, Virology & Host Pathogen Interaction
**The EMBO Journal (2021) 40: e107985**

## Introduction

Pandemics represent a major threat to global health. The coronavirus SARS-CoV-2 has given rise to one of the worst still ongoing pandemics in recent history, COVID-19. As of June 2021, the virus had infected more than 181 million individuals and caused almost four million deaths globally. The regulatory agencies approved vaccines with unprecedented speed, but their availability remains limited, particularly in low-income countries. Moreover, the observation of repeated infections within one year (Dao *et al*, 2020; Huang *et al*, 2020) suggests that not all individuals develop a protective immune response and that not everyone will respond sufficiently and sustainably to the vaccinations either. The situation worsened with the occurrence of even more virulent and transmissible strains, such as Alpha/UK/B.1.1.7 carrying the strain-charactering N501Y Spike mutation or the Beta/South African strain B.1.351. This raises the need for the continued development of efficient therapies and vaccines with long-lasting efficacy to combat COVID-19. More broadly, preparedness to rapidly respond to any newly emerging infectious disease or active crisis is essential for limiting potentially devastating consequences to worldwide health and the global economy.

Therapeutic approaches that interfere with SARS-CoV-2 genome replication, e.g., using the nucleoside analog Remdesivir, showed only moderate if any efficacy in clinical trials so far (Beigel *et al*, 2020; Spinner *et al*, 2020). One reason is the inefficient inhibition of the viral RNA polymerase (Kokic *et al*, 2021) but perhaps also that the compound cannot prevent the initial infection of cells. Therefore, it would be desirable to neutralize SARS-CoV-2 before it can enter and infect cells.

Infection is initiated when the Receptor-Binding Domain (RBD) of the SARS-CoV-2 Spike docks to angiotensin-converting enzyme 2

1 Department of Cellular Logistics, Max Planck Institute for Biophysical Chemistry, Göttingen, Germany
2 Institute of Molecular Oncology, GZMB, University Medical Center, Göttingen, Germany
3 Department of Molecular Biology, Max Planck Institute for Biophysical Chemistry, Göttingen, Germany
4 Animal facility, Max Planck Institute for Biophysical Chemistry, Göttingen, Germany
5 Institute of Medical Microbiology and Virology, University Medical Center, Göttingen, Germany
*Corresponding author. Tel: +49 551 3960757; E-mail: mdobbel@uni-goettingen.de
**Corresponding author. Tel: +49 551 2012400; E-mail: goerlich@mpibpc.mpg.de
†These authors contributed equally to this work

(ACE2), a plasma membrane protein of the target cell. This interaction is a prerequisite for the cleavage of the Spike protein by TMPRSS2 and for subsequent fusion of viral and cellular membranes, and it is thus required for viral entry into cells (Lan *et al*, 2020; Wang *et al*, 2020; Yan *et al*, 2020; Zhou *et al*, 2020a). Therefore, preventing the interaction of the RBD and ACE2 represents a promising strategy for the therapy and prophylaxis of COVID-19.

Vaccination to raise antibodies against the Spike is the most widely used measure for blocking virus entry (Dai & Gao, 2020; Dong *et al*, 2020b). It might, however, take two vaccinations and thus up to several weeks before a sufficient protective antibody level has built up. In contrast, passive immunization can be effective immediately. The serum of convalescent patients (Liu *et al*, 2020), monoclonal antibodies (DeFrancesco, 2020; Andreano *et al*, 2021), or decoy receptors (Chan *et al*, 2020) provided protection in patients or at least in model systems.

A formidable challenge for the therapeutic antibody approach has been the emergence of several virus mutations that alter RBD epitopes such that the virus can escape neutralization. A particularly negative impact can be attributed to the E484K/Q, K417N/T, and the L452R mutations. These mutations can occur singly, e.g., in the Epsilon/Californian B.1.429 strain (L452R), or in combination, for example, in the Gamma/Brazilian P.1 strain (K417T, E484K, N501Y), in the Beta/South African B.1.351 strain (K417N, E484K, N501Y) or the Indian B.1.617 lineage (L452R and E484Q in the Kappa/B.1.617.1 and B.1.617.3 variants, or L452R and T478K in the Delta/B.1.617.2 sub-lineage). These strains are linked to severe outbreaks with very high infection rates and a reduced efficacy of vaccines (Wibmer *et al*, 2021; Zhou *et al*, 2021).

Therapeutic anti-SARS-CoV-2 monoclonal IgGs are typically administered in gram amounts (Chen *et al*, 2020). As they can only be manufactured in mammalian cells, it already is a challenge to produce enough material to treat even just a fraction of the patients who need therapy.

Single-domain VHH antibodies (also called nanobodies) are derived from camelid heavy-chain-only antibodies, whose antigen-binding sites are composed of just one peptide chain (Hamers-Casterman *et al*, 1993). This makes their coding regions straightforward to clone from cDNA (without combinatorial issues) into phage display vectors for subsequent selection of high-affinity binders. Nanobodies have been used for a wide range of applications (Ingram *et al*, 2018; Cheloha *et al*, 2020), and their production in *E. coli* or yeast is potentially less expensive and more scalable than conventional antibody manufacturing. VHHs against SARS-CoV-2 (in various forms) have been described recently (Custódio *et al*, 2020; Esparza *et al*, 2020; Hanke *et al*, 2020; Huo *et al*, 2020; Schoof *et al*, 2020; Wrapp *et al*, 2020a; Xiang *et al*, 2020; Koenig *et al*, 2021; Pymm *et al*, 2021; Xu *et al*, 2021), so far mostly with moderate (nM) monomer affinities and/or still limited thermal stability.

Here we report the development of anti-RBD VHH monomers that completely neutralize the infectivity of SARS-CoV-2 at concentrations as low as 17 pM (0.2 μg/l). These include leads that are hyperthermostable to ≥ 95°C, either from the beginning or following structure-guided engineering. We further describe the use of collagen fragment fusions with these VHHs to obtain trimers that match the symmetry of the Spike and neutralize the virus even at a concentration of 1.7 pM (0.6 pM trimer) or 40 nanograms per liter. Moreover, we constructed nanobody tandems and identified nanobody monomers that each avidly bind an RBD with an extreme combination of escape mutations (K417T, E484K, N501Y, L452R) and neutralize also the Beta/South African virus strain B.1.351 at low-picomolar concentrations. Finally, we discovered that certain VHHs enforce native folding of the RBD even in the *E. coli* cytosol (where proper RBD-folding fails otherwise) without obstructing the primary epitope for neutralization. Such bacterially expressed immunogen raises the perspective of inexpensive and yet effective vaccines that can be rapidly adapted to newly emerging viral strains.

## Results and Discussion

### Anti-RBD and anti-S1ΔRBD VHH antibodies

This endeavor has aimed at obtaining VHH antibodies that neutralize SARS-CoV-2 potently. For this, we targeted the S1 fragment of the SARS-CoV-2 Spike protein (Wrapp *et al*, 2020b), which contains two main globular domains: a large N-terminal, lectin-like domain (residues 27–290) and a receptor-binding domain (RBD, residues ~ 334–527). The RBD mediates the interaction with the host cell receptor ACE2 and is thus essential for infectivity.

We immunized three alpacas five times in weekly intervals with the complete S1 fragment and the RBD, collected blood samples four days after the last immunization, isolated RNA from lymphocytes, amplified VHH-coding regions by nested RT–PCRs, and cloned them as cDNAs into a phagemid to yield three separate immune libraries with complexities of around $10^9$ independent clones each. These libraries were subjected to phage display, with baits being either the RBD (yielding anti-RBD VHHs) or the S1 fragment with competition by an excess of free RBD (to obtain "anti-S1ΔRBD" VHH antibodies that recognize S1 epitopes outside the RBD).

672 selected clones were sequenced and classified by sequence similarity (Fig 1A). 60 representatives of all VHH classes were then recombinantly produced, purified, and characterized by a range of assays. Nanobodies with two disulfide bonds ($n = 10$) were expressed in the periplasm of *E. coli*.

Nanobodies containing just a single structural disulfide bond ($n = 50$) were initially expressed in the cytoplasm of the *E. coli* SHuffle Express strain (NEB). As a first validation, we labeled them with fluorophore-maleimides (Pleiner *et al*, 2015) through ectopic cysteines (introduced at the N and C-termini). Subsequent immunofluorescence on (SARS-CoV-2 Spike-) transfected HeLa cells revealed that most of them (~ 90%) indeed recognized the SARS-CoV-2 Spike protein.

For the neutralization assays described below, we also needed to detect newly synthesized Spike protein within infected cells. For safety reasons, these samples had to be fixed for a long time (1 h) in 4% paraformaldehyde (PFA). Thus, only VHHs with highly fixation-resistant epitopes were expected to function. Nevertheless, we identified numerous VHHs that stained the viral Spike protein brightly and specifically as judged by the absence of signal in non-infected cells (Fig 1B and C). These included the seven anti-RBD nanobodies shown in Fig 1B as well as a set of anti-S1ΔRBD VHHs (Fig 1C), which target the S1 domain at some epitope outside the RBD.

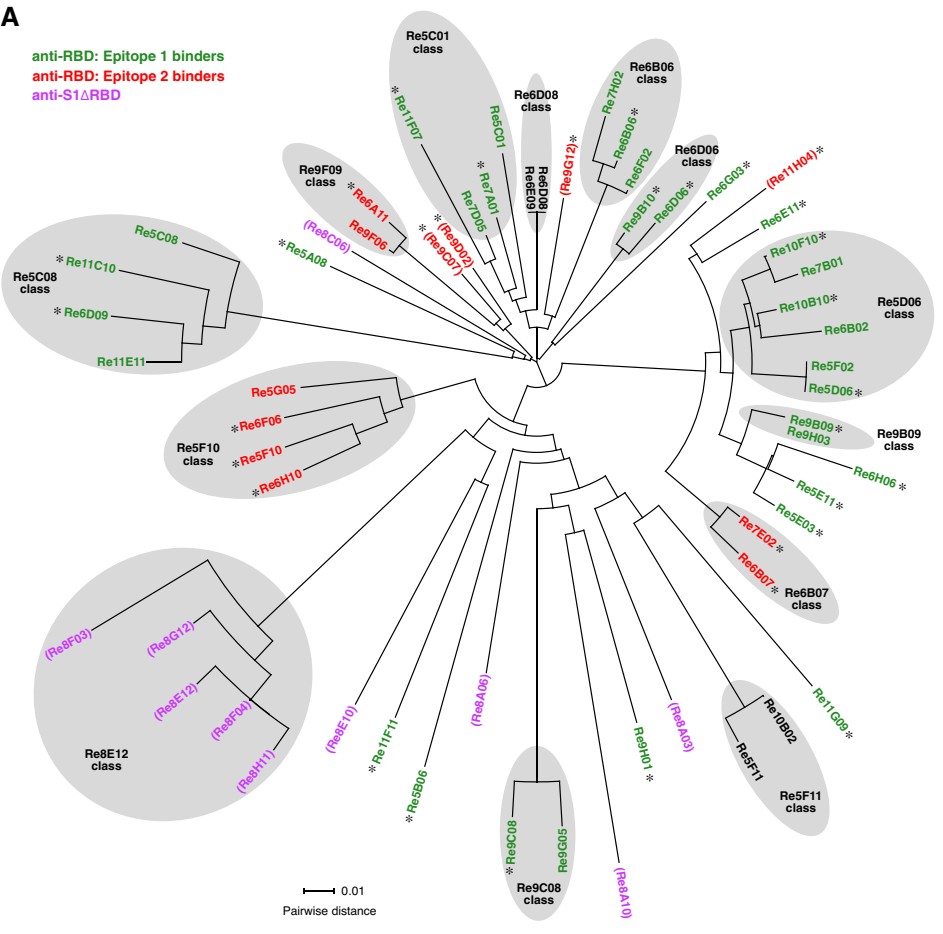

**A**

anti-RBD: Epitope 1 binders
anti-RBD: Epitope 2 binders
anti-S1ΔRBD

0.01
Pairwise distance

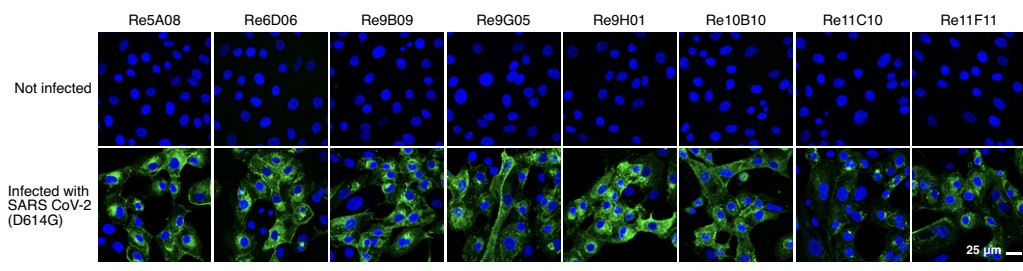

**B**

**Immunofluorescence labeling of Vero E6 cells**
**Detection of: DNA and SARS-CoV-2 Spike protein**

anti-RBD VHH (AF488-labeled)

Re5A08 | Re6D06 | Re9B09 | Re9G05 | Re9H01 | Re10B10 | Re11C10 | Re11F11

Not infected

Infected with SARS CoV-2 (D614G)

25 µm

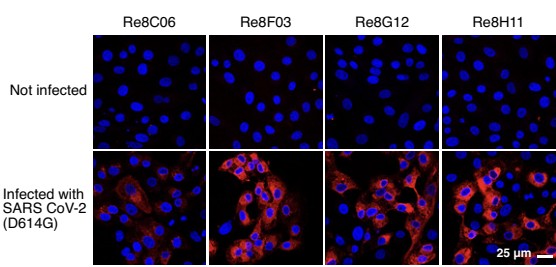

**C**

**Immunofluorescence labeling of Vero E6 cells**
**Detection of: DNA and SARS-CoV-2 Spike protein**

anti-S1ΔRBD VHH (AF568-labeled)

Re8C06 | Re8F03 | Re8G12 | Re8H11

Not infected

Infected with SARS CoV-2 (D614G)

25 µm

**Figure 1.**

◀

**Figure 1. Phylogenetic relationship of the anti-SARS-CoV-2 Spike VHH antibodies identified and characterized in this study.**

A The sequences of the isolated VHH antibodies (49 anti-RBD and 10 anti-S1ΔRBD nanobodies) were aligned using Clustal Omega (Sievers *et al*, 2011). The circular phylogram was reconstructed with Dendroscope (Huson & Scornavacca, 2012). Where applicable, sequence classes are indicated. The anti-RBD nanobodies are colored according to the two RBD epitopes identified in Fig 5 (with epitope 1 in green, epitope 2 in red, and asterisks marking nanobodies included in Fig 5). Anti-S1ΔRBD nanobodies are colored in magenta. VHH antibodies that did not neutralize SARS-CoV-2 are shown in parentheses. See main text and Fig 5 for details.

B Vero E6 cells were inoculated with SARS-CoV-2, fixed with paraformaldehyde after two days, stained with 30 nM of the indicated Alexa Fluor 488 (AF488)-labeled anti-RBD VHH antibodies, and imaged by confocal laser scanning microscopy (CLSM).

C Immunofluorescence (IF) staining as in B, but with the indicated AF568-labeled anti-S1ΔRBD VHH antibodies.

## Low-picomolar RBD affinities

The RBD affinity of a neutralizing nanobody determines the fraction of Spike molecules that this nanobody can mask at a given concentration. Affinity thus is a major determinant of neutralization power (see below). Therefore, we analyzed the VHH·RBD interactions by bio-layer interferometry (BLI; Abdiche *et al*, 2008). To this end, we immobilized the RBD (through a biotinylated Avi-tag; Beckett *et al*, 1999) to streptavidin sensors and measured the association and dissociation of free nanobodies supplied in a range of concentrations (Fig 2A). Alternatively, we immobilized VHHs (labeled with two $PEG_{11}$-biotins) to the sensors and measured the binding of the free RBD (Fig 2B). This setup is more robust for discerning extremely slow dissociation kinetics.

In this assay, the previously described VHH-72 (Wrapp *et al*, 2020a) bound the RBD with a dissociation constant ($K_D$) of ~ 27 nM (Fig 2A), which agrees well with the earlier reported $K_D$ of ~ 39 nM. The newly selected VHHs Re6B06 and Re9F06 bound the RBD with intermediate affinity, i.e., with $K_D$ values of 12 and 4 nM, respectively (Fig 2A).

For several nanobodies, we observed an extremely tight RBD-binding (Fig 2B). These belong to distinct classes and include Re5D06 ($K_D$ ~ 2 pM), Re9B09 ($K_D \leq 1$ pM), and Re6H06 ($K_D \leq 1$ pM). For the latter two, dissociation remained non-detectable even after very long dissociation times. Note that BLI is not reliable at discerning single-digit picomolar or even femtomolar affinities. However, these numbers are consistent with our observation that those nanobodies resisted stringent, overnight off-rate selections by phage display. Further low-picomolar RBD binders include Re5F10 (~ 30 pM), Re9H01 (~ 10 pM), as well as another Re9B09 class member, namely, Re9H03 (~ 25 pM).

## VHH antibodies with excellent SARS-CoV-2 neutralization potency

To assess if the obtained VHHs would prevent viral entry, we set up a stringent SARS-CoV-2 neutralization assay based on Vero E6 cells. This cell line is very susceptible to viral infection, because it expresses high levels of the SARS-CoV-2 receptor ACE2 (Li *et al*, 2003) and is deficient in its interferon response (Emeny & Morgan, 1979). For inoculation, we employed a patient-derived SARS-CoV-2 isolate that carries the infection-enhancing D614G mutation of the Spike (Stegmann *et al*, 2021; preprint: Zhang *et al*, 2020).

To measure infection and virus neutralization, we employed three independent readouts: first, the cytopathic effect (CPE) caused by the virus; second, quantitative RT–PCR to measure the release of newly replicated viral RNA; and third, immunofluorescence with anti-RBD and anti-S1ΔRBD VHH antibodies to detect newly synthesized viral Spike proteins inside infected cells. The latter assay turned out to be the most sensitive one as it can detect a single infected cell per well, even at an early stage of infection.

Fig 3A and B demonstrates two of the neutralization readouts for the example of VHH-72 (Wrapp *et al*, 2020a), where SARS-CoV-2 was pre-incubated with serial VHH dilutions before adding the virus to cultured cells. Two days later, infection (respectively virus neutralization) was scored to determine the lowest VHH concentration that still reliably neutralized the virus completely (IC99$^+$ values). Without the addition of virus, no viral components were detected, neither by IF nor by RT–PCR. By contrast, viral infection for 48 h led to bright intracellular Spike signals and an approximately 10,000-fold higher viral RNA load than in the inoculum. Pre-incubation of the virus with ≥ 500 nM VHH-72 (~ 7 mg/l) nearly completely blocked infection, while lower concentrations had no effect. This number is similar to the previous report on VHH-72 and roughly 10- to 20-fold higher than the $K_D$ for this nanobody (Fig 2A; Wrapp *et al*, 2020a). This suggests that neutralization requires that the majority (perhaps ≥ 90%) of RBDs on a viral particle are blocked.

To identify neutralizers among the newly isolated anti-RBD VHHs, we tested them initially at a concentration of 500 nM (7.5 mg/l) and found that 43 out of 60 prevented infection completely. By contrast, nanobodies binding the S1 domain outside the RBD (belonging to the Re8 series) were inactive in neutralization.

An important criterion in pharmaceutical development is the lowest concentration at which an antiviral antibody is still effective. Indeed, it makes a great difference if 0.7- 7 grams (as in the case of the anti-SARS-CoV-2 antibody Bamlanivimab; Chen *et al*, 2020) or one milligram would be required as an effective therapeutic dose. This determines the cost per treatment and, if many patients need to be treated, it matters how many therapeutic doses can be produced at all. Likewise, adverse side effects by the antibody itself or by contaminants will scale with the applied dose. Thus, it is highly desirable to obtain VHH antibodies that effectively block viral infection at the lowest possible concentration.

We therefore retested our anti-RBD VHHs repertoire for neutralization potency. This revealed that numerous nanobodies of intermediate RBD affinity neutralize down to the low nM-range (Fig 4A and B, Appendix Fig S1A and B). These include Re9F06 (17 nM), Re5F10 (5 nM), Re6B07 (5 nM), and Re6B06 (50 nM, see below). Strikingly, we found that the high-affinity RBD binders are sub-nanomolar neutralizers (see Fig 4C and D, Appendix Fig S1C and D): Re9B09, Re9H01, and Re9H03 block SARS-CoV-2 infection already at 167 pM, whereas Re5D06 and Re6H06 neutralize even down to 50 pM, which corresponds to ~ 0.7 µg VHH antibody per liter. When injected into a patient of 70 kg body mass, and assuming dilution into the extracellular fluid

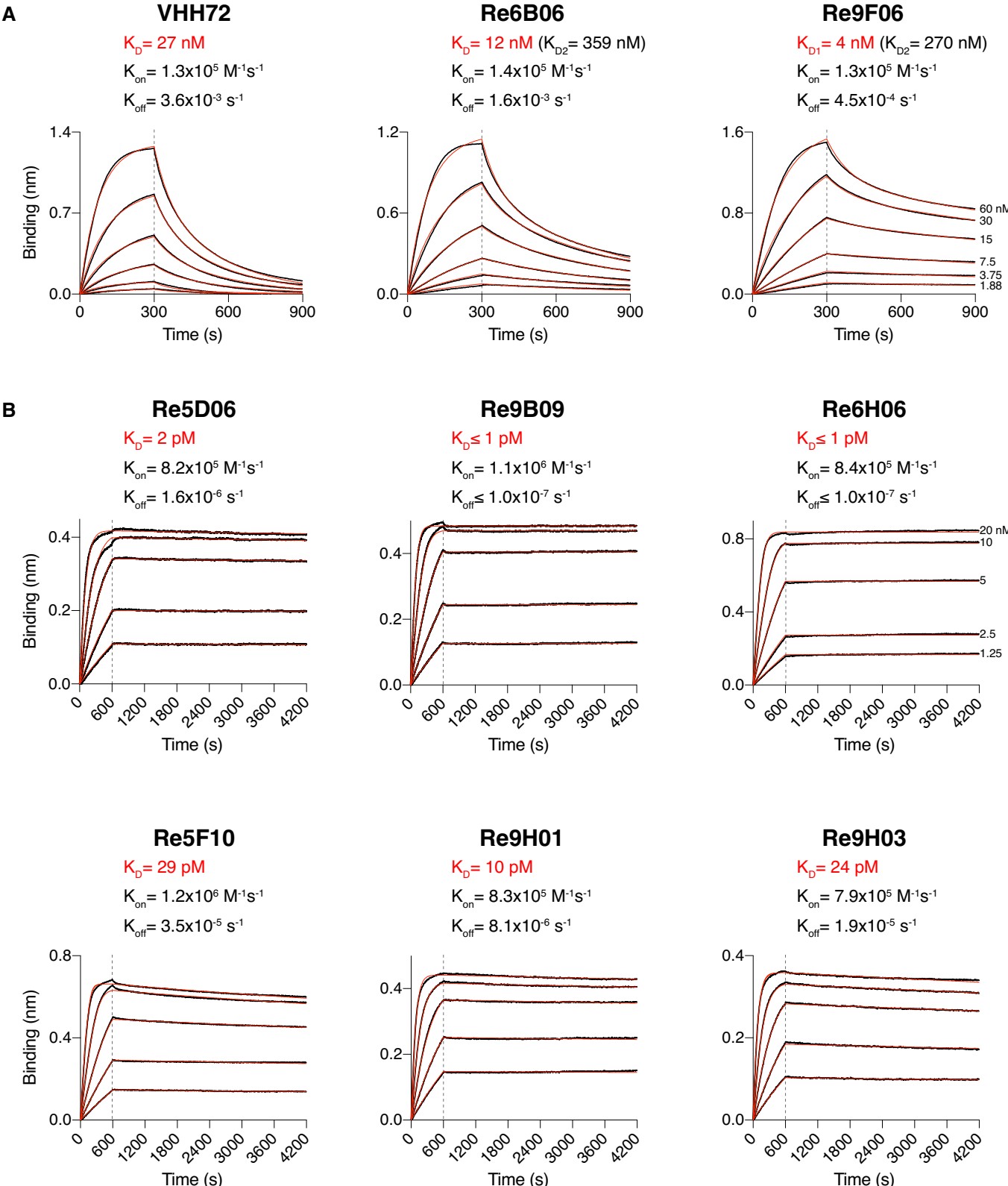

**Figure 2.**

**Figure 2.   Affinities of VHH antibodies for the RBD, determined by bio-layer interferometry (BLI).**

BLI sensorgrams of intermediate (A) and high-affinity (B) VHHs.

A   SARS-CoV-2 RBD-loaded sensors were incubated with nanobodies at the indicated concentrations for 300 s, followed by dipping in assay buffer for 600 s. The response curves (black) were fitted using either a 1:1 (for VHH-72) or 2:1 (for Re6B06 and Re9F06) binding models to obtain $K_D$s. Fitted curves are shown in red. For Re6B06 and Re9F06, some secondary binding with weaker affinity at high VHH concentrations was observed (denoted as $K_{D2}$).

B   Nanobody-loaded sensors were dipped into wells containing RBD at the indicated concentrations for 600 s and then incubated with assay buffer for 1 h. The response curves (black) were fitted using a mass transport model. Fitted curves are shown in red.

**A**

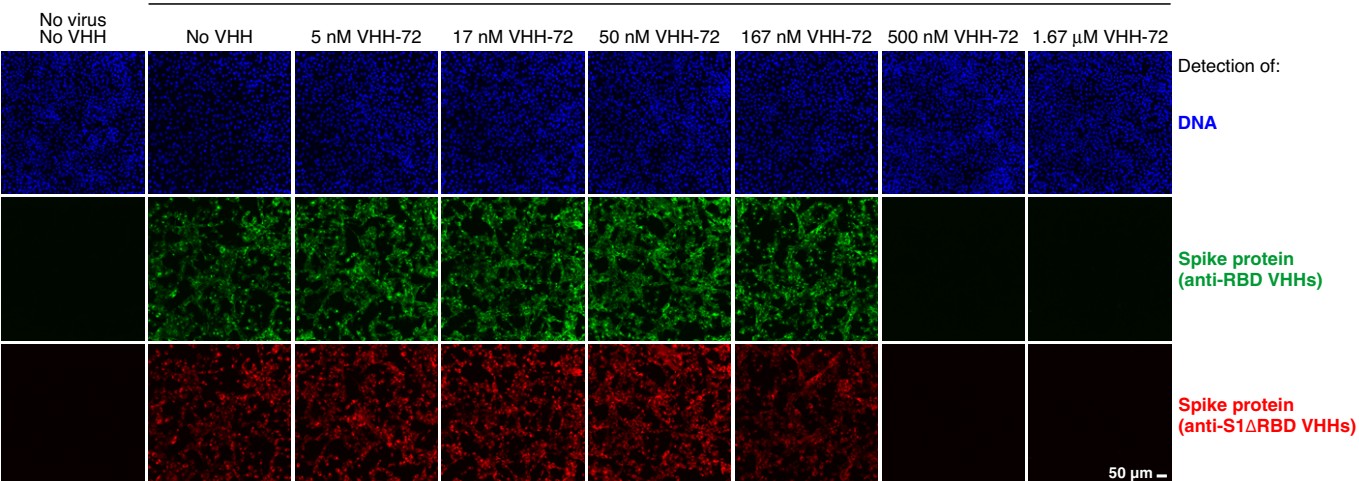

**B**

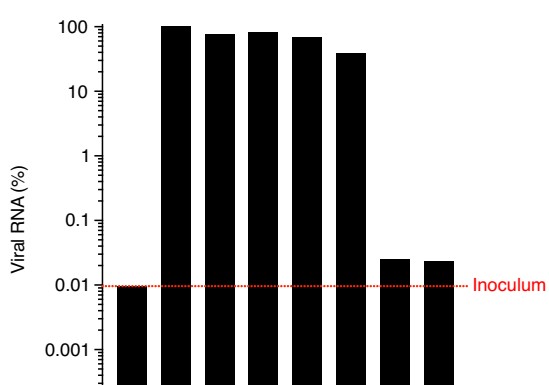

**Figure 3.   Fluorescence-based assay to assess the neutralization potency of VHH antibodies.**

A   Vero E6 cells were infected with SARS-CoV-2, pre-incubated with indicated concentrations of the neutralizing VHH-72 (Wrapp *et al*, 2020a). Cells were fixed two days after inoculation, stained with sets of anti-RBD (green) and anti-S1ΔRBD (red) nanobodies (see Fig 1B and C), and analyzed by CLSM. We used a cocktail of such fluorophore-labeled VHH antibodies to ensure that negative fluorescence readings truly indicated neutralization and thus absence of viral infection and not just masking of the IF epitope by the tested nanobody. See Appendix Fig S1 for independent biological replicates.

B   Vero E6 cells were infected as in A. Viral RNA in the culture supernatants was quantified by reverse transcription (RT)–qPCR. RNA signals are normalized to the respective minus-VHH controls (100%). Note the log10 scale of the y-axis. "Inoculum" (red dotted line) marks the RNA level detected at the time of infection. See also Appendix Fig S1.

with a volume of 14 l with negligible elimination, then one milligram VHH Re5D06 should (theoretically) already be sufficient to exceed the expected therapeutically effective concentration by a factor of 100. These VHHs and their further enhanced derivatives (see below) are now excellent candidates for treating SARS-CoV-2-infected patients and eventually open new avenues for preventive strategies.

**The RBD contains two preferred VHH epitopes**

The surface of the RBD would be sufficiently large to accommodate several VHH antibodies at a time. Indeed, staining of Spike-expressing HeLa cells with fluorescent nanobodies revealed that Re10B10 (a Re5D06 sequence class member) was able to bind simultaneously with Re7E02 (Fig 5A): The Re7E02 signal was outcompeted only with unlabeled Re7E02 but not with Re10B10, while the Re10B10 stain was outcompeted only with unlabeled Re10B10, but not with Re7E02 (Fig 5A). Thus, the two nanobodies occupy non-overlapping sites on the RBD and represent "compatible binders".

We tested anti-RBD nanobodies of all classes and found that they all competed with either the Re5D06 class member or with Re7E02 (for a selection, see Fig 5A). In the following, we will refer to the

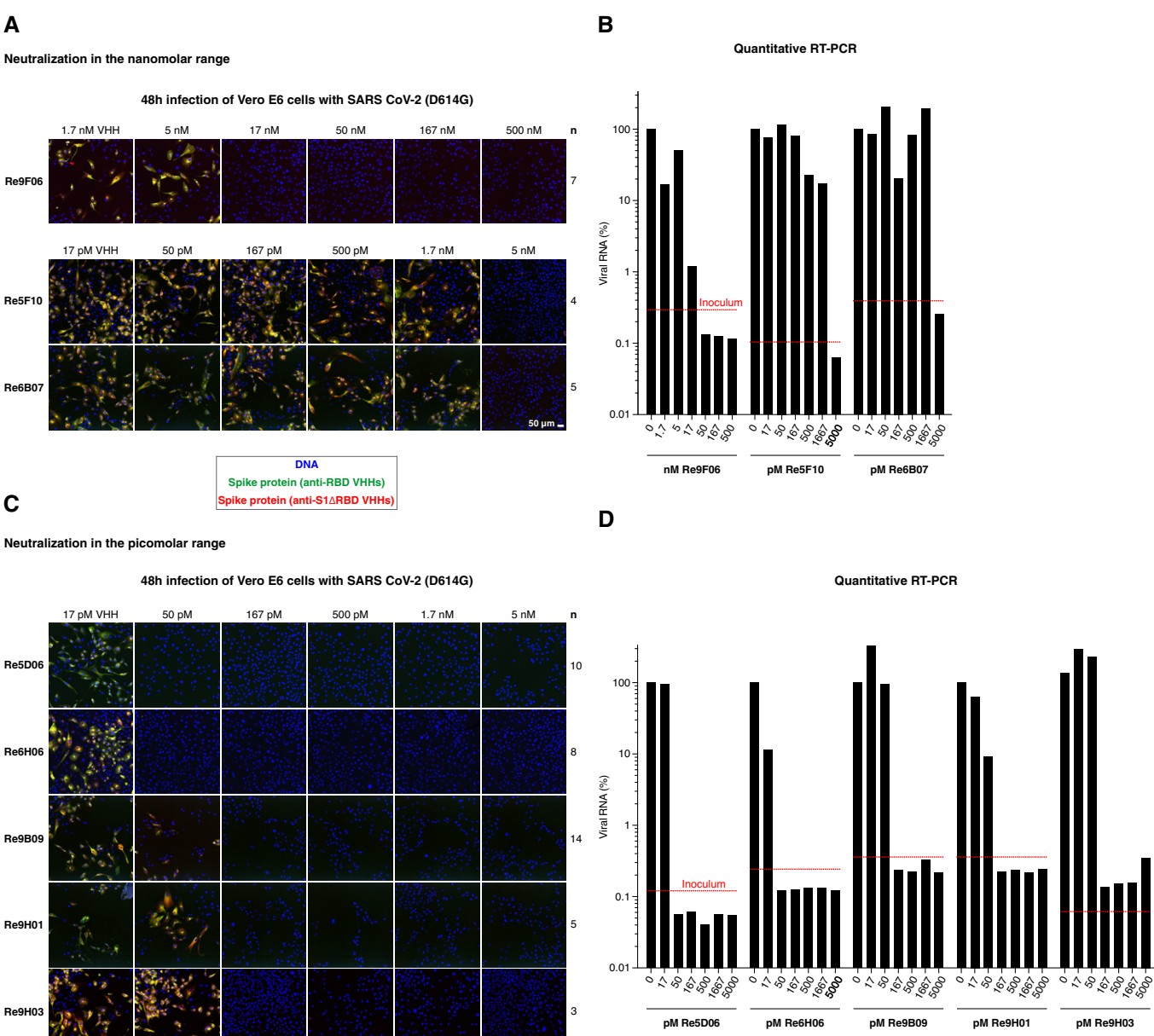

**Figure 4. Neutralization of SARS-CoV-2 by VHH antibodies.**

A–D    Neutralization of SARS-CoV-2 by the indicated VHH antibodies at nanomolar (A and B) or picomolar (C and D) concentrations. The neutralization experiment was performed as described in Fig 3A and B. "*n*" indicates the number of independent biological replicates (see also Appendix Fig S1).

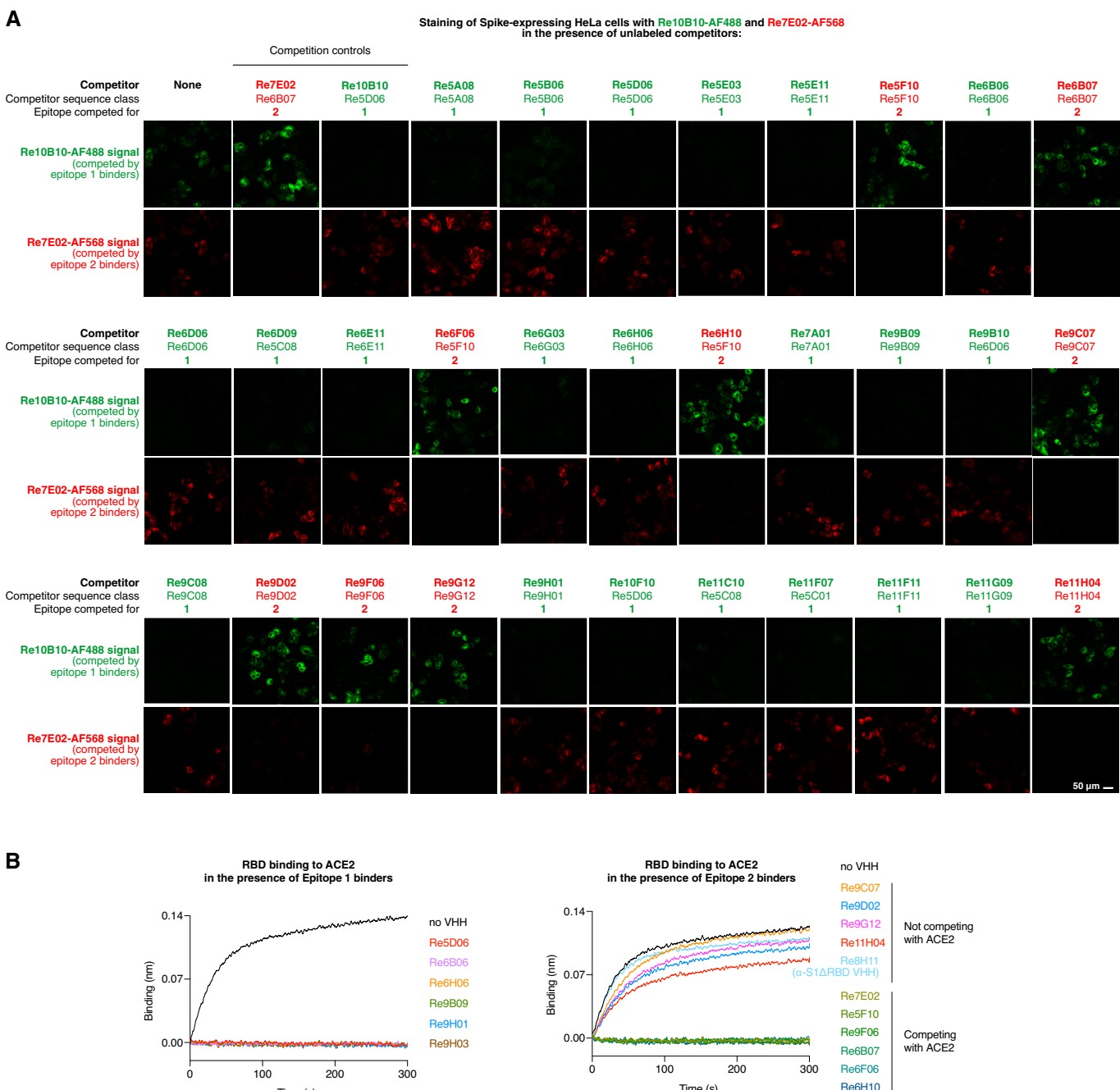

**Figure 5. Epitope-binning with the indicated VHH antibodies.**

A  HeLa cells were transiently transfected to express the SARS-CoV-2 Spike protein. Following fixation, cells were stained for 1 h with fluorophore-labeled Re10B10 (5 nM, green) and Re7E02 (15 nM, red) in the presence of the indicated unlabeled VHH competitors. Competitor (150 nM) was added 20 min prior to the labeled nanobodies. The weakly binding competitors Re5A08, Re9F06, Re6B06, and Re6D06 were added as trimers (see below). Cells were imaged by CLSM. For each competitor, the sequence class and the binding site on the RBD (epitope 1 or 2) are indicated.

B  ACE2 competition experiments with selected VHHs from A. 50 nM RBD was mixed with indicated nanobodies (at 500 nM). Binding to ACE2 (immobilized on the sensors) was monitored by BLI.

Re5D06 epitope as "epitope 1" and to the Re7E02 epitope as "epitope 2".

A BLI assay (Fig 5B), measuring the association of the RBD with ACE2, revealed that all epitope 1-binders (e.g., Re5D06, Re6H06, or Re9B09) blocked the ACE2·RBD interaction. This explains their neutralizing effect. Some of the epitope 2-binders (e.g., Re5F10, Re7E02, or Re9F06) also competed with ACE2. Other epitope 2-binders (e.g., Re9C07 or Re9G12), however, did not compete

(Fig 5B). This suggests that epitope 2 is rather peripheral with respect to the ACE2 footprint.

## Fold-promoting nanobodies

To gain insights into the structural basis of super-neutralization, we were interested in crystallizing RBD·VHH complexes. For this, we needed large amounts of the respective components. Obtaining the nanobodies was rather trivial. The RBD, however, failed to fold when expressed in *E. coli*, even when the NEB SHuffle Express strain (which allows disulfide-bond formation in the cytoplasm) was used (Bessette *et al*, 1999). In fact, the small soluble RBD fraction obtained in these preparations was not recognizable by the super-neutralizing VHH Re5D06, but instead largely trapped by the chaperone GroEL (Appendix Fig S2). This is in line with previous reports of failed folding of coronaviral RBDs in *E. coli* (Chen *et al*, 2005; preprint: Jegouic *et al*, 2020).

We then tried an unconventional approach and tested if co-expressing specific nanobodies together with the RBD would help. None of the tested VHHs recognizing epitope 1 (i.e., the Re5D06 epitope; Fig 5A) had the desired positive effect. However, each and every nanobody targeting epitope 2 (the epitope of Re7E02; Fig 5A and B) counteracted the non-productive association with GroEL, improved solubility, and formed stable complexes with the RBD. Re9F06 is an example, and Appendix Fig S3A shows that Re9F06 exhibits its fold-promoting effect also as a fusion with the RBD. In this setting, we observed nearly complete solubility of the RBD and yields in the order of ~ 10 mg/l *E. coli* shaking flask culture, even without any optimization. Furthermore, the fusion bound the compatible VHH Re5D06 (Appendix Fig S3B), indicating a correct fold of the RBD.

For structural analysis, we next prepared a ternary complex of the RBD with the fold-promoter Re9F06 and the potently neutralizing Re5D06 by co-expressing all three entities (Appendix Fig S4A). The purified complex crystallized (under *in situ* proteolysis conditions; Appendix Fig S4B–D) in space group 4 ($P12_1$1) with two complexes per asymmetric unit, but diffracted only to a limited resolution of 3.3 Å. Optimizing construct boundaries, crystallization conditions, and additives eventually led to crystals in space group 19 ($P2_12_12_1$) that diffracted up to 1.5 Å resolution. The structure was solved by molecular replacement, using the known structures of the SARS-CoV-2 RBD (PDB 6YZ5; Huo *et al*, 2020) and nanobodies with deleted CDR loops as initial search models, and refined to a resolution of 1.75 Å with good statistics (Appendix Table S1). The structure shows one Re9F06·RBD·Re5D06 complex per asymmetric unit (Fig 6A and Appendix Table S1).

The RBD itself was virtually identical to previous reports (e.g., PDB 6YZ5 Huo *et al*, 2020), including correctly formed disulfide bonds (Fig 6A), validating the above-described approach of nanobody-assisted folding. The two VHHs were bound to almost opposite sides of the RBD (Fig 6A), yet, both would clash with RBD-bound ACE2 (Fig 6B, see below). The clash with the fold-promoter Re9F06 is small but sufficient to prevent docking to ACE2, explaining its neutralizing activity (Fig 4A). The clash is not caused by an overlap of the RBD-binding sites of VHH and ACE2, but rather by an overlap between the VHH scaffold and its CDR2 loop with ACE2 (compare Fig 6A with 6B).

The fold-promoting Re9F06 contacts a conformational epitope (Fig 6D and Appendix Fig S5). It thus "reads" and stabilizes the

RBD fold. The fold-promoting effect itself, however, is a different quality. We presume that the nanobody captures a folding intermediate and prevents (by conformational selection) a diversion of the RBD to a non-productive folding path. An additional effect might be that Re9F06 sterically excludes trapping of the RBD by GroEL. The VHH thus acts as a (non-releasable) chaperone—a concept that had not been appreciated before in nanobody biology.

These observations immediately suggest practical applications, like, e.g., employing *E. coli* as the most economical expression system to produce RBD-based immunogens. This could lead to more affordable vaccinations against SARS-CoV-2 and possibly other coronaviruses, in particular for low-income countries. Furthermore, for adapting a vaccine candidate to new escape mutants, it will be more straightforward to modify just an *E. coli* expression plasmid for immunogen production than to re-engineer a mammalian production cell line for the same purpose.

The presence of the fold-promoting nanobody as an RBD-ligand might provide additional benefits. Re9F06 leaves the main epitope for neutralization fully exposed; nevertheless, it prevents ACE2 from masking this epitope (Fig 6B) and should thereby improve its presentation to the immune system. This might also reduce the side effects of the immunization caused by an undesired binding of an RBD-vaccine to ACE2-presenting target cells with subsequent antibody-binding and thus opsonization of such cells.

## Structural basis for highly potent neutralization

As mentioned above, the highly potent VHH Re5D06 clashes extensively with ACE2; in fact, it uses a very similar binding site on the RBD (Fig 6A and B). In addition, the structure indicates a remarkably tight interaction between this nanobody and the RBD, which is consistent with its impressive affinity for the RBD (Fig 2B). The buried interface is extensive (~ 2,135 Å$^2$) and features perfect shape-complementarity between the nanobody and its target (see Fig 6E, left panel). It includes salt bridges (direct and water-mediated), cation-$\pi$ interactions, hydrogen bonds, planar and T-shaped $\pi$ stacking interactions, and other hydrophobic contacts (Appendix Fig S6). The interactions encompass the long CDR3 loop, CDR1, as well as residues of the nanobody scaffold, such as R50 and R52 (Fig 6E). On the RBD side, seven residues are particularly noteworthy. These include four tyrosines (Y449, Y453, Y489, Y505) and three phenylalanines (F456, F486, F490), suggesting that the nanobody covers a rather "sticky" area of the RBD.

The long CDR3 loop greatly contributes to the shape complementarity with the RBD surface. Binding would be entropically disfavored if, in the non-bound state, this loop was flexible or adopted a different conformation. To address this issue, we also determined the crystal structure of free Re5D06 to 1.25 Å resolution (Fig 6F, Appendix Table S1). We observed that the free CDR3 loop conformation was nearly identical with the RBD-bound form (RMSD of 0.96 Å over backbone atoms of residues 99–115; Fig 6F)—differences were only evident for a few side chains. This pre-adoption of the "bound-loop conformation" minimizes the entropic penalty for binding and provides an additional explanation for this nanobody's extraordinary neutralization power.

Remarkably, CDR3 comprises three highly exposed tyrosines critical for RBD-binding (Fig 6F), although they should have a high propensity for getting buried. This suggests a strong stabilization of

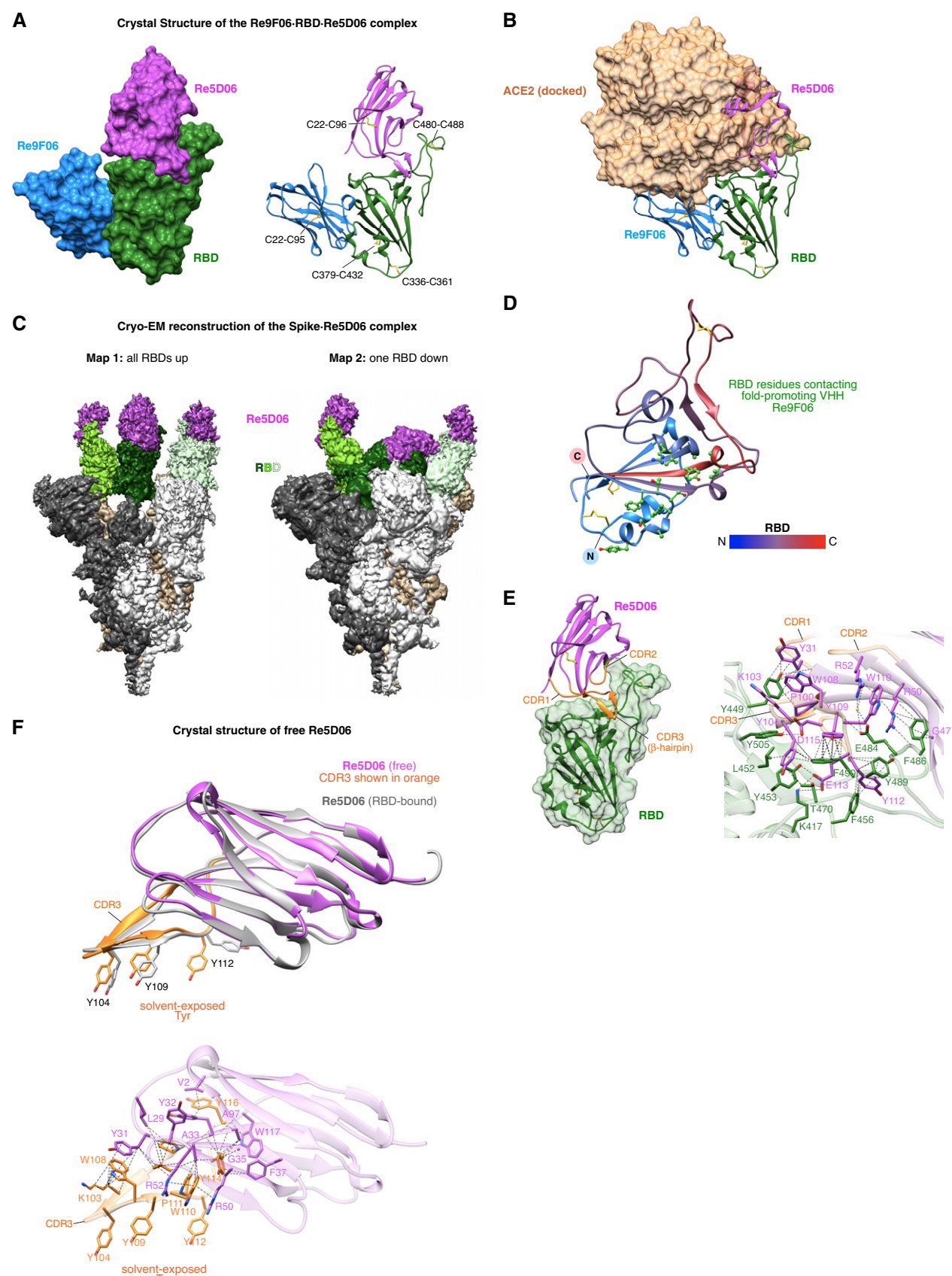

**Figure 6.**

**Figure 6.   Structural characterization of the ternary Re9F06·RBD·Re5D06 complex and free Re5D06.**

A   Crystal structure of the Re9F06·RBD·Re5D06 complex (at 1.75 Å resolution; Appendix Table S1) as surface (left) or ribbon representation (right). The intramolecular disulfide bonds (shown as yellow sticks) are labeled.

B   Ribbon representation of the Re9F06·RBD·Re5D06 complex as in A, but with RBD-bound ACE2 shown as a semitransparent brown surface. Docking of ACE2 is based on the alignment of the RBD with the SARS-CoV-2 RBD·ACE2 complex (PDB ID 7KMS (Zhou *et al*, 2020b); RMSD = 0.986 Å). Note the clash of ACE2 with Re5D06 and Re9F06.

C   Cryo-EM reconstructions of the Spike·Re5D06 complex (Appendix Table S2). SARS-CoV-2 Spike (Hsieh *et al*, 2020) was incubated with an excess of Re5D06, purified by size exclusion chromatography, and vitrified immediately for structural characterization by cryo-EM (Appendix Fig S8). The two conformational classes detected were refined to high resolution (2.8 Å global). The sharpened maps are shown at low contour levels. The protomers of the homotrimeric Spike are shown in white, tan or gray, respectively, and are further color-coded as indicated. See Appendix Fig S8–S11 for details and more information.

D   The SARS-CoV-2 RBD from A is shown as a ribbon, colored according to the indicated color gradient, with its N-terminus in blue and C-terminus in red. Disulfide bridges are depicted as yellow sticks. Re9F06 has been omitted for clarity. RBD side chains that interact with the fold-promoting VHH Re9F06 are shown as green "ball-and-sticks". See Appendix Fig S5 for a comprehensive analysis of the Re9F06·RBD interface.

E   Molecular details of the RBD·Re5D06 interaction. Left: overview of the RBD·Re5D06 complex, with the RBD shown as a green ribbon overlayed with its semitransparent surface. Re5D06 is shown as a ribbon (in magenta) with orange CDR loops. Right: Details of the RBD·Re5D06 interaction interface. RBD and Re5D06 are shown as semitransparent ribbons colored as on the left, with selected interface side chains depicted in green (RBD) or magenta (Re5D06). Blue marks nitrogen, oxygen is shown in red. A water molecule is shown as a yellow sphere. Dashed lines link interacting atoms (distance ≤ 4 Å). Lines pointing onto backbones indicate contacts to carbonyl-carbons or amide groups. See Appendix Fig S6 for more details.

F   Crystal structure of free Re5D06, solved to 1.25 Å resolution (Appendix Table S1). Top: free Re5D06 is shown in magenta, with CDR3 (featuring three solvent-exposed tyrosine residues) colored in orange. For comparison, Re5D06 in its RBD-bound conformation is overlayed (RMSD = 0.986 Å) in gray. Bottom: intramolecular interactions that stabilize CDR3. CDR3 side chains are depicted in orange; all other side chains are displayed in magenta. See E for further details. Extensive interactions in free Re5D06 stabilize a CDR3 conformation (with solvent-exposed Tyr 104, 109, 112) that requires only relatively minor structural changes for strong RBD binding.

the loop conformation. Indeed, a closer inspection of the structure revealed such an extensive stabilization by the hydrogen bonds of the β-hairpin and by other hydrogen bonds and hydrophobic, π-stacking, and cation-π contacts (Fig 6F).

A final aspect is that Re5D06 binding is compatible with all reported RBD "up" and "down" conformations of the complete Spike homotrimer (Appendix Fig S7). Indeed, our cryo-EM analysis of the Spike·Re5D06 complex revealed two classes, both of which show robust occupancy of all RBDs, regardless of their relative conformation (Fig 6C; Appendix Fig S8–S11). This is undoubtedly an advantage in terms of the antiviral effect because all conformations are immediately susceptible to neutralization.

## Hyper-thermostable anti-RBD VHHs with low-picomolar neutralization potency

For the intended therapeutic application, the anti-SARS-CoV-2 VHH antibodies should not only be highly potent in virus neutralization but also "developable". This includes sufficient stability for withstanding a lengthy, large-scale production process as well as transportation and storage (ideally for several years in liquid formulation) without denaturation and loss of activity. Among numerous parameters, thermostability is perhaps the single best predictor for developability (Goldberg *et al*, 2011; Jarasch *et al*, 2015), the rationale and experience being that stability at high temperatures also translates to superior stability at body (37°C), ambient (20°C), and storage temperatures (4°C). Thermostability to 75°C is usually considered the threshold for antibody developability.

We started our analysis with extreme conditions and incubated our leads at 90°C for 5 min (at 1 μM concentration). After cooling and removing possibly formed aggregates by centrifugation, RBD binding was assessed by BLI, which is well suited to detect any loss of active nanobody. Interestingly, there was no difference between heated and untreated samples of either Re9F06, Re5F10, Re9B09, or Re5D06 (Fig 7A and B; see below for other hyperthermostable nanobodies). Thus, Re9B09 and Re5D06 (epitope 1-binders) as well

as Re9F06 and Re5F10 (epitope 2-binders) are either hyperthermostable or can robustly refold to their native states following heat treatment.

Further analysis of bacterially expressed Re5D06 by differential scanning fluorometry (DSF) revealed, however, an onset of melting at about 50°C (Fig 7C). This nanobody had been produced in the SHuffle Express strain, which is engineered for cytoplasmic disulfide-bonding. The formation of the structural disulfide bond remained, however, incomplete. This could explain the still rather low melting point. Indeed, secretory production in *Pichia pastoris* (which allows for quantitative disulfide-bonding) shifted the onset of unfolding to 65°C. Such thermostability already exceeds that of most human or *E. coli* proteins but appears still rather low for a prospective therapeutic candidate.

The crystal structure of the RBD·Re5D06 complex revealed a potentially destabilizing cavity within the nanobody's hydrophobic core (Fig 7D). The cavity hosts the hydrophobic portion of the used crystallization additive (benzyl dimethyl ammonium propane sulfonate; Appendix Fig S12A). In turn, this suggested that the packing of the hydrophobic core could be improved and that this might stabilize the nanobody further. To this end, we used Rosetta-modeling (Leaver-Fay *et al*, 2011) and identified I34 M, S49A, and V79Y as potentially core-stabilizing mutations (Fig 7D). Visual inspection of the structure suggested three more mutations (Q39E, N77D, K87E) that should introduce additional, stabilizing salt bridges (Appendix Fig S12B).

We tested a total of > 30 variants and found two seemingly optimal solutions: Re5D06R15 (R15) and R28 (Appendix Fig S12B). R15 has two additional exchanges: S53N (found in other selected Re5D06 class members) and V93D for improving surface packing and surface polarity. R28 is further fold-stabilized by A24I, I27S, I51W, and S54N exchanges (Appendix Fig S12B). In addition, it carries a Y109H substitution in CDR3. Remarkably, both R15 and R28 resisted melting at 95°C (Fig 7E) but still achieved a complete neutralization (IC99[+]) down to monomer concentrations of 17–50 pM (0.2–0.7 μg per liter; Fig 7F). As of now, this is an

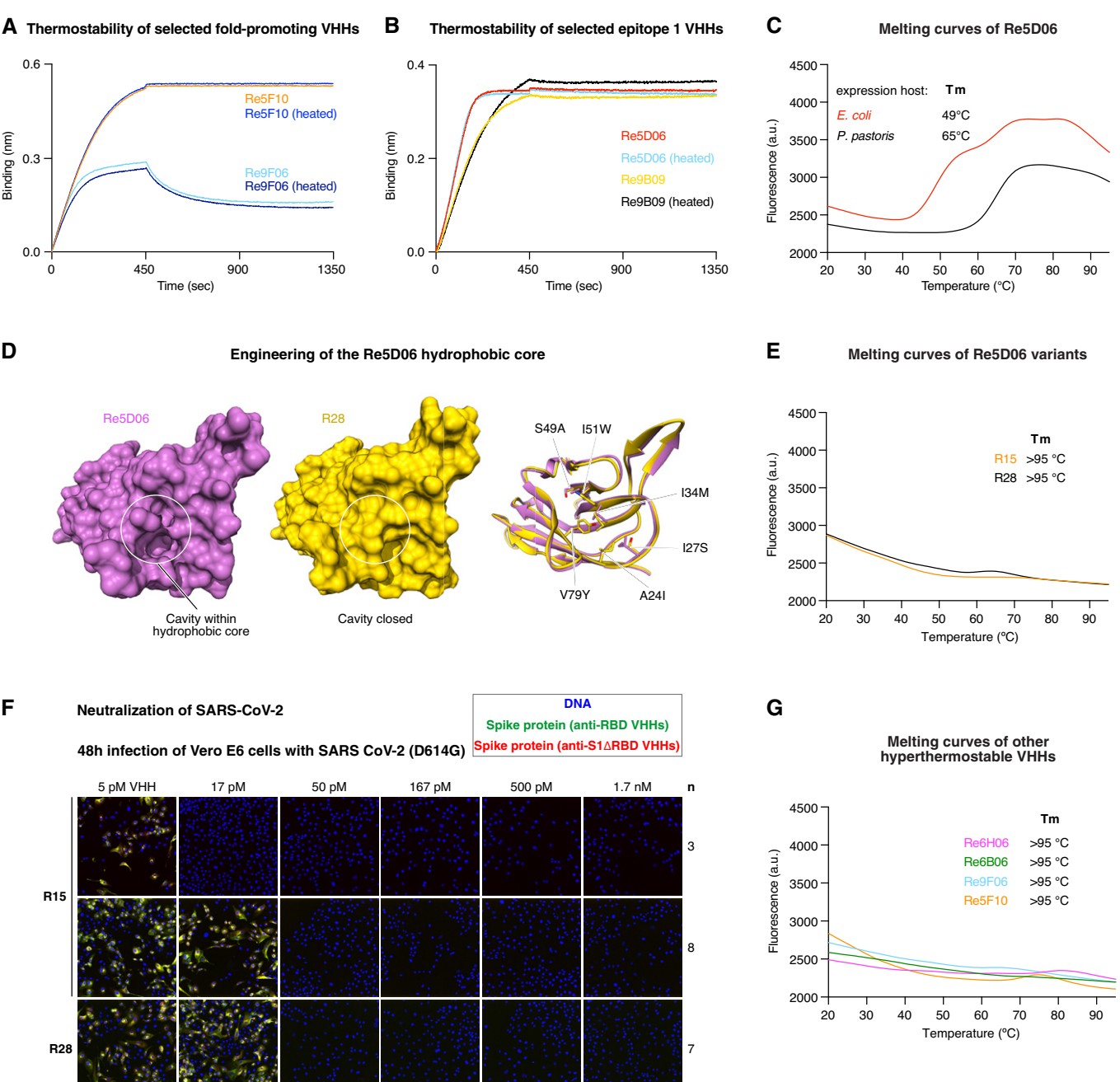

**Figure 7. Thermal characterization of selected neutralizing anti-RBD VHHs.**

A, B  Indicated VHH antibodies (1 µM) were incubated at room temperature or at 90°C for 5 min and centrifuged for 20 min at 20,000 *g*. The supernatants were diluted 50-fold (20 nM) and analyzed for RBD-binding by BLI.

C  Re5D06 produced in the indicated expression host was analyzed by differential scanning fluorimetry (DSF) with a temperature range from 20 to 95°C (in the presence of SYBR Orange). Unfolding is measured as an increase in fluorescence (caused by the exposure of hydrophobic residues that bind the dye). Melting temperatures are defined as the inflection point of the melting curve before reaching the first melting peak. Stable proteins produce no melting peak.

D  The left two panels show surface representations of Re5D06 (magenta) and the thermostable variant R28 (yellow). The ribbon diagram (right) depicts an overlay of both nanobodies with changed hydrophobic core side chains shown as sticks. Note that these mutations close a large cavity originally present in Re5D06. See Appendix Fig S12B for an overview of the engineered Re5D06 variants shown here.

E  Thermostability of the Re5D06 variants R15 and R28, measured as in C.

F  Neutralization of SARS-CoV-2 by the thermostable Re5D06 variants R15 and R28 at the indicated concentrations. The neutralization experiment was performed as described in Fig 3A.

G  DSF measurements of additional hyperthermostable VHHs as in C.

unprecedented combination of extraordinary antiviral potency and hyperthermostability (for benchmarks of stability, see Hussack *et al*, 2011; Kunz *et al*, 2017).

In addition, we identified several VHH antibodies that were hyperthermostable even before any optimization. These include the epitope 2-binders Re9F06 and Re5F10 (Fig 7A and G), whereby the latter is stabilized by an additional disulfide between CDR3 and the VHH scaffold. The main epitope 1-binders Re6B06 and Re6H06 are hyperthermostable as well (Fig 7G), with Re6H06 being again stabilized by an additional disulfide bond (between CDR2 and 3). As described above (Fig 2B and 4C), Re6H06 is also one of the strongest RBD binders ($K_D \leq 1$ pM) and most potent neutralizers (at 50 pM).

### Symmetry-matching in Spike neutralization

Re6B06, however, bound the RBD with only a ~ 12 nM $K_D$ (Fig 2A) and neutralized SARS-CoV-2 in the 50 nM range (Fig 8C). To overcome such limitations for moderately binding nanobodies, we devised a strategy that exploits avidity effects. As discussed below, this strategy promises additional benefits, including an improved plasma half-life of applied nanobodies.

The traditional approach for gaining avidity would be to fuse the monovalent VHH antibody to the Fc part of a human IgG and make the VHH bivalent (Wrapp *et al*, 2020a). This could impose a far stronger binding by avidity effects, but only if the valency and the distance of target sites fit the geometry of the antibody. This indeed appears problematic in our case, because a dimeric IgG might well dock to two of the RBD molecules of a homotrimeric Spike, but this would leave the third one in an unbound state.

We reasoned that homo-trimerization might be a better strategy for gaining avidity. This requires a trimeric fusion partner with special properties: First, for use in patients, it should be as non-immunogenic as possible. This excludes bacteriophage-derived moieties, such as the T4 foldon (Letarov *et al*, 1999) or trimerizing versions of non-human leucine-zippers. Instead, it should be derived from an abundant, extracellular, human protein. Second, it should be small for the sake of material economy; yet, the resulting fusion should exceed the size-limit for fast renal clearance of ~ 65 kDa. Third, it should lack N-glycosylation sites (which would not be used in *E. coli* and modified in a non-human manner in yeast). Fourth, it should be stable and trimerize already at picomolar or even lower concentrations. These restraints left us with two candidates—the NC1 trimerization domains of human collagens XV (PDB 3N3F, 54 residues; Wirz *et al*, 2011) and collagen XVIII (PDB 3HSH; 54 residues; Boudko *et al*, 2009).

These collagen NC1 modules have been used for trimerizing VHHs before (Sánchez-Arévalo Lobo *et al*, 2006; Cuesta *et al*, 2009; Alvarez-Cienfuegos *et al*, 2016), but not for symmetry-matching and targeting homotrimeric targets. We produced > 30 different NC1-VHH fusions in *E. coli* and obtained all of them in high yields, purity, and stability (see Fig 8A for an example). We compared several designs and found the best neutralization potency for VHH-spacer-collagen XVIII NC1 fusions, with a spacer that was ~ 30 residues long, flexible, and negatively charged. Trimerization *per se* was not the only determinant of neutralization efficacy, as the NC1 trimerization domain of collagen XVIII performed consistently better than analogous collagen XV variants, regardless of the type of fusion (N- or C-terminal) or the linker used.

Figure 8C shows a proof-of-principle, namely, for the rather weak RBD-binder VHH-72 (Wrapp *et al*, 2020a). The VHH-72 monomer neutralized down to 500 nM (7 mg/l). In contrast, its collagen XVIII trimer still neutralized at 50 pM (17 pM trimers or 1.2 µg/l). This is an increase in potency by a factor of 10,000 (based on molarity) or 6,000 (based on mass). For comparison, a dimerization by an Fc-fusion achieved only in a ~ 10-fold gain in potency (Wrapp *et al*, 2020a).

The trimerization also lowered the minimal neutralizing concentration of fold-promoting nanobody Re9F06 by a very large factor, namely, from 50 nM down to 167 pM. Nanobodies that already neutralize potently as monomers benefited less from the trimerization. The trimer of Re6D06, for example, neutralized only 100-fold better (at 17 pM) than the corresponding monomer (1.7 nM; Fig 8C) and the super-neutralizing VHH Re5D06 showed no more than a 3-fold improvement (from 50 to 17 pM; Fig 8C). This smaller gain can be explained by avidity effects not accelerating association but only impeding dissociation. If the dissociation is already very slow, then a further slowdown will not have much benefit. In other words, neutralization by Re5D06 appears limited by its on-rate but no longer by its already very high affinity.

We evaluated numerous VHHs to identify the best one for the trimerized format. Counterintuitively, this was not the most potent monomer, but Re6B06, whose monomer neutralized SARS-CoV-2 only down to 50 nM (Fig 8C). Its trimerized version, however, was 30 000-fold more potent and neutralized still at 1.7 pM (referring to the VHH moiety), corresponding to 0.6 pM trimer or 40 ng VHH fusion per liter (Fig 8C), consistent with a great increase in avidity due to trimerization (Fig 8B). This Re6B06-spacer-ColXVIII fusion thus outperformed all other variants tested by us or by others (Custódio *et al*, 2020; Dong *et al*, 2020a; Esparza *et al*, 2020; Hanke *et al*, 2020; Huo *et al*, 2020; Moliner-Morro *et al*, 2020; Schoof *et al*, 2020; Wrapp *et al*, 2020a; Xiang *et al*, 2020; Koenig *et al*, 2021; Pymm *et al*, 2021; Xu *et al*, 2021). It is now a serendipitous coincidence that the Re6B06 class also features the greatest possible thermostability and thus combines two of the most desired features. This example, however, also illustrates that not only concept and design matter, but also unforeseen details.

### Nanobody tandems that tolerate current escape mutations

By now, several SARS-CoV-2 strains with mutated RBDs have emerged that are causing devastating outbreaks. These include the Alpha/UK B.1.1.7 variant with an N501Y mutation in the RBD, the Beta/South African B.1.351 variant (K417N, E484K, N501Y), Gamma/Brazilian P.1 (K417T, E484K, N501Y), Iota/New York City B.1.526 (E484K), Epsilon/Californian B.1.429 (L452R), and the Indian B.1.617 lineage (with L452R and E484Q in the Kappa/B.1.617.1 and B.1.617.3 variants, or L452R and T478K in the Delta/B.1.617.2 sub-lineage).

The emergence of such mutant strains is presumably driven by the selective pressure of the immune system, particularly in immunocompromised, persistently infected patients (Clark *et al*, 2021; Starr *et al*, 2021). Escape mutants can bypass antibody-based immunity in previously infected or vaccinated individuals. This has the potential of sustaining infection waves in populations that have acquired herd immunity against earlier strains.

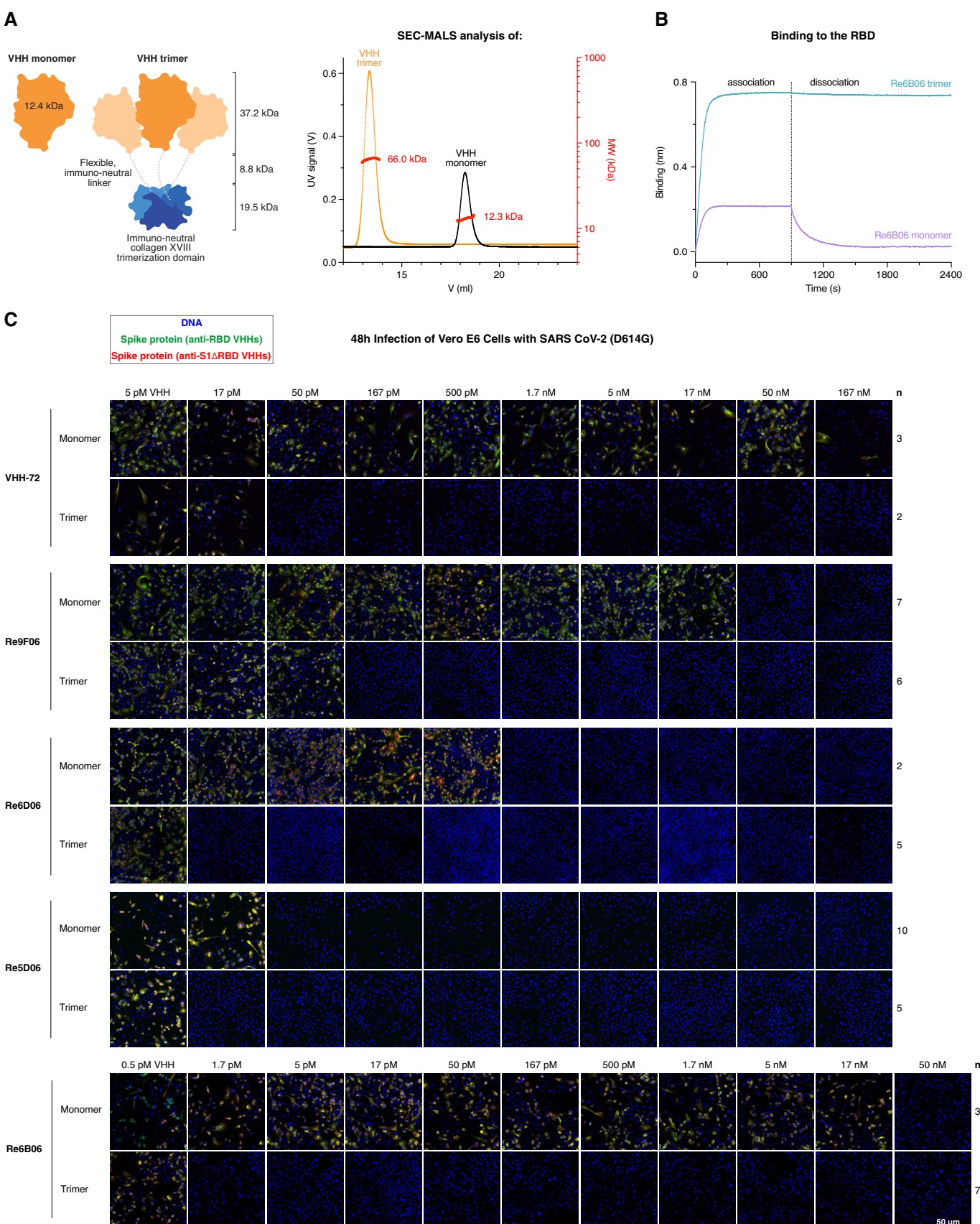

**Figure 8.**

**Figure 8.  Neutralization of SARS-CoV-2 with symmetry-matched VHH antibodies.**

A   Left: Schematic representation of VHH antibodies produced in *E. coli* as monomers or homo-trimeric fusions with the human collagen XVIII trimerization module. The VHH trimers match the threefold rotational symmetry of the SARS-CoV-2 Spike protein. The molecular weight of the indicated modules is shown. Right: As an example, monomeric and trimeric forms of Re7H02 (a Re6B06 class member, Fig 1A) were analyzed by size exclusion chromatography (SEC, Superdex 200 column) coupled with multi-angle light scattering (MALS). The elution profiles (UV absorbance signal), as well as the calculated molecular weights (per slice), are plotted, and the average molecular weight is indicated.

B   Re6B06 monomer and trimer variants (20 nM) were analyzed by BLI for RBD-binding, with 900 s association and 1,500 s dissociation steps. Note the avidity effect of Re6B06 trimerization.

C   Neutralization of SARS-CoV-2 by the indicated VHH antibodies. The neutralization experiment was performed as described in Fig 3A. Note the gain in neutralization efficacy when a VHH is trimerized. Re6B06, for example, is 30,000-fold more potent when symmetry-matched with the virus Spike.

Escape mutants also pose a tremendous challenge for therapeutic antibodies, which may be rendered ineffective by a given mutation. Indeed, several monoclonal anti-SARS-CoV-2 antibodies developed for treating COVID-19 are ineffective against the Beta/South African and Gamma/Brazilian variants (Garcia-Beltran *et al*, 2021; Hoffmann *et al*, 2021; Li *et al*, 2021; preprint: Tada *et al*, 2021; Wibmer *et al*, 2021; Zhou *et al*, 2021).

Previously reported SARS-CoV-2-neutralizing nanobodies also appear affected by such mutations if they target the major epitope 1. For example, the high-affinity nanobody Nb20 ($K_D \sim 1$ pM; PDB 7JVB; Xiang *et al*, 2020) contacts both E484 and L452. An E484K RBD mutation cannot be accommodated into the structure and would probably abolish binding. The L452R mutation would also be detrimental. Likewise, Nb6 ($K_D$ 450 pM; PDB 7KKK; Schoof *et al*, 2020) would be severely hit by the E484K mutation. The same applies to NbH11-D4 and NbH11-H4 ($K_D$s, ~ 10 nM; PDB 6YZ5 and 6ZH9; Huo *et al*, 2020). Finally, the RBD-binding of VHH-E ($K_D \sim 2$ nM; PDB 7KN5; Koenig *et al*, 2021) will also be impeded by a loss of charge complementarity (E484K) or loss of hydrophobic contacts, steric hindrance, and unfavorable side chain-packing (L452R). This emphasizes that those escape mutations center at highly antigenic sites.

Given the relevance of the now circulating RBD mutations, we decided to explore their impact on our VHH leads. Our Re5D06·RBD model can accommodate the N501Y exchange without clashes or structural rearrangements. Indeed, the mutation had little effect on the binding of Re5D06 or of any other tested nanobody (Fig 9A and B).

The combined Beta/South African (K417N, E484K, N501Y) or Gamma/Brazilian (K417T, E484K, N501Y) mutations, however, weakened the Re5D06·RBD interaction to 0.1–0.5 nM $K_D$s (Fig 9A). This affinity decrease can be explained by the Re5D06·RBD structure, namely, by E484 making a water-bridged ionic bond to R52 of the nanobody (Fig 6E, Appendix Fig S6), which is lost in the E484K mutant. The K417N/T exchange causes the loss of a peripheral salt bridge to E113 of the nanobody (Fig 6E, Appendix Fig S6). The Californian L452R mutation reduced affinity to ~ 400 pM (Fig 9A). Re5D06$^{Y104}$ contacts L452 (Appendix Fig S6), and the newly introduced arginine is likely to pack non-favorably against Re5D06$^{Y104}$. Attempts to restore high-affinity binding by compensatory mutations failed and instead compromised the folding of the nanobody. We therefore sought alternative ways to identify nanobodies with high affinity to all SARS-CoV-2 variants of concern.

As a first approach, we exploited that Re9F06 and Re5D06 can simultaneously bind to the same RBD molecule (see Fig 5A) and that a tandem of the two will bind more avidly than either nanobody alone. Re9F06 is hyperthermostable (Fig 7G) and docks to the

"fold promoter" epitope ("epitope 2") that is unaffected by those escape mutations. We fused Re9F06 through a structure-optimized linker to R28—the hyperthermostable Re5D06 variant described above. The resulting tandem indeed bound the Beta/South African, Gamma/Brazilian, and Epsilon/Californian variants rapidly and with hardly detectable dissociation (Fig 9A). We expect this tandem fusion to bind the RBD of all Indian/B.1.617 sub-lineages with similar affinity, because it likely also tolerates the less stringent E484Q mutation of the Kappa and B.1.617.3 sub-lineages as well as T478K (which is not part of the Re5D06/R28·RBD interface) of the Delta variant. For a more stringent test, we produced a quadruple (K417T, L452R, E484K, N501Y) RBD mutant that combines the Gamma/Brazilian and Epsilon/Californian mutations. The Re9F06-R28 tandem bound this extremely mutated RBD remarkably well, with only slow dissociation (Fig 9C). The analogous Re9F06-Re9B09 tandem showed even less dissociation from the Beta/South African and Gamma/Brazilian RBD mutants (Fig 9A) and captured the quadruple RBD essentially irreversibly (Fig 9C).

### Nanobody monomers with exceptional mutation tolerance

To find yet another solution to the mutation problem, we re-selected our immune libraries (after the initial four rounds of anti-wild type RBD selection) with the above-mentioned quadruple RBD mutant. Sequencing of recovered clones identified a simplified pattern of nanobody classes. The Re5F10 class became very prevalent. It is hyperthermostable (Fig 7A), competes the ACE2·RBD interaction (Fig 5B), and binds with 30 pM affinity to the fold-promoter epitope ("epitope 2"; Fig 5A) of either the wild type or the quadruple mutant RBD (Figs 2B and 9C). Re6H06 was selected at a lower frequency, but BLI revealed remarkable mutation resistance and ≤ 10 pM binding to either the Beta/South African or the Gamma/Brazilian as well as 50 pM binding to the Epsilon/Californian RBD mutant (Fig 9B).

Finally, we observed a very strong selection for Re9B09 class members with an E115K exchange at CDR3, exemplified by Re9H03. This class targets the main epitope 1 (Fig 5A) and belongs to the most potent neutralizers (Fig 4C and D). Remarkably, Re9H03 bound essentially irreversibly to the Gamma/Brazilian, the Beta/South African as well as to the quadruple mutant (100-fold better than Re9B09 itself; Fig 9B and C), whereas some dissociation from the Epsilon/Californian single L452R mutant was clearly notable. This indicates that some of the other escape mutations (probably K417N/T) improve nanobody binding. As the alpacas were immunized only with the "wild type" RBD, this indicates that their immune system had already generated a diversity of antibodies that "anticipated" even the worst combination of escape mutants.

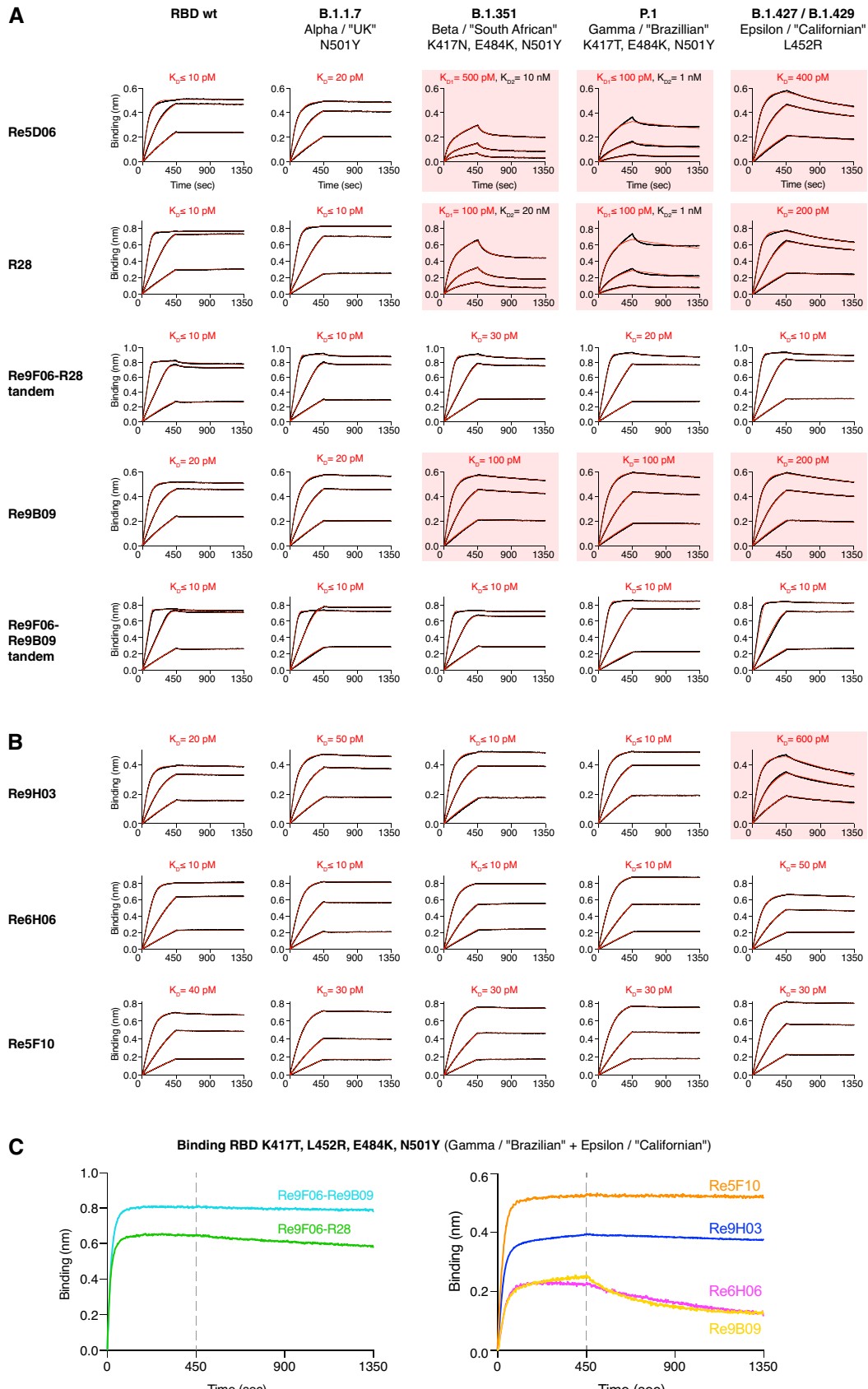

**Figure 9.**

**Figure 9. Characterization of VHH antibodies that tolerate major RBD escape mutations.**

A, B  The indicated nanobody constructs were immobilized on BLI sensors and dipped into solutions containing 3-fold dilutions of WT or mutant RBDs (20, 6.66, and 2.22 nM) for 450 s. RBD dissociation in assay buffer was followed for 900 s. The response curves with biphasic dissociations were fitted using a 2:1 model. All other curves were fitted using a mass transport model. RBDs with impaired VHH binding are highlighted with red boxes.

C  Binding of nanobodies to the quadruple RBD mutant was monitored by BLI. Immobilized RBD was incubated with 100 nM of nanobodies for 450 s, followed by assay buffer for 900 s.

Data information: We use the simplified SARS-CoV-2 nomenclature proposed by the World Health Organization (WHO), but also provide the PANGO lineage identifier and the obsolete naming of variants by their geographic origin, because all of these naming conventions have been used in the literature cited in this work.

## Highly potent neutralization of the Beta/South African SARS-CoV-2 strain

In a final set of experiments, we assessed the neutralization potency of our mutation-tolerant nanobodies, using the Beta/South African B.1.351 variant as a representative of the recently emerged mutant strains. As we rely on nanobody-based detection of newly made viral proteins, we had to ensure that the mutant Spike also yields an unambiguous signal. This was achieved with a mix of nanobodies against epitopes outside the RBD (anti-S1ΔRBD), against the non-mutated fold-promoter epitope (Re7E02 or Re9C07), and by the mutation-tolerant main epitope-binder Re9H03 (Appendix Fig S13A). By contrast, Re6D06, which fails in mutant RBD binding, also failed to stain mutant Spikes of infected cells (Appendix Fig S13A and B), suggesting that combinations of mutation-sensitive and mutation-tolerant nanobodies can diagnose virus variants by simple staining procedures.

We then compared three strategies for the actual neutralization of B.1.351 (Fig 10). We first tested the most promising VHH monomers and observed potent neutralization by Re5F10 (at 1.7 nM), Re6H06 (at 170 pM), Re9B09 (at 1.7 nM), and by the mutant-preferring Re9B09 class member Re9H03 (at 50–170 pM; Fig 10A). Even more potent B.1.351 neutralization was evident by the tandems Re9F06-R28 (50 pM), Re9F06-Re9B09 (50 pM), and Re9F06-Re6H06 (17 pM; Fig 10B). The collagen XVIII trimer of Re9F06 showed good neutralization even at 17 pM concentration (corresponding to 5.8 pM trimers; Fig 10B) and still 170 pM when a fivefold higher virus titer was used for infection.

For an anticipated clinical use, the nanobodies described here (see Appendix Tables S3 and S4 for an overview) have the advantages of extraordinary stability and suitability for economic production. Their small size comes with fast renal excretion, but their plasma half-life can be prolonged by the above-mentioned strategies of protein fusion and oligomerization. Fusing nanobodies to an Fc-fragment might well boost their antiviral efficacy (e.g., through opsonization and subsequent elimination of virus-infected cells); however, human-type glycosylation of Fc fragments requires cumbersome and expensive expression in mammalian cells. In several respects, the collagen XVIII trimerization domain appears to be an excellent alternative fusion partner. It confers far better avidity

toward Spike homotrimers than bivalent IgGs, expresses well in bacteria or yeast and is devoid of N-glycosylation sites. Likewise, all nanobody modules and deployed spacers lack N-glycosylation sites.

The RBD epitope 2 was, so far, spared from escape mutations. This can be explained by structural and functional constraints, i.e., by Spikes poorly tolerating changes at this position. This epitope might also be under lower escape pressure because it makes only a minor contribution to the neutralizing immune response in humans. In any case, the current conservation predicts that VHHs against this epitope will tolerate future RBD escape mutations better than the main epitope-binders. In fact, this is a strong argument for basing therapeutic homotrimers on epitope 2-binders, and from this perspective, the collagen XVIII-trimerized, hyperthermostable VHH Re9F06 is a seemingly perfect clinical candidate.

Given that nanobodies against the main epitope neutralized more potently than epitope 2-binders, they should also be deployed for COVID-19 therapy. Re6H06, Re9B09, and the Re5D06-derived R28 are excellent candidates because they feature extraordinary potency, high mutation tolerance (Re9B09, Re6H06) and hyperthermostability (Re6H06, R28). Here, tandem fusions with Re9F06 are the perfect format as Re9F06 confers very robust tolerance toward current and probably also toward future escape mutations. We currently pursue the clinical development of these tandems for therapy and prophylaxis, either applied by route of injection or inhalation.

In addition, we need to be prepared for future pandemic events. Although our alpacas had been immunized with only the RBD from the canonical SARS-CoV-2 strain, it was straightforward to isolate mutation-adapted VHH variants simply by another round of phage display with a mutant RBD bait. One can now imagine repeating such a reselection whenever novel mutant strains escape the current nanobody formats. Moreover, even when further expansion of the anti-SARS-CoV-2 antibody repertoire will be needed, the already vaccinated animals can be reimmunized with the relevant RBD mutants. In contrast to a new immunization project, this will probably require no more than just one additional boost and can therefore be regarded as a fast-track for generating further sets of broadly neutralizing nanobodies. Fortunately, alpacas are long-lived if cherished and well looked after, so we are confident that our immunized alpacas can help tackle future strains of SARS-CoV-2 and thus contribute to our preparedness.

**Figure 10. Neutralization of SARS-CoV-2 B.1.351 by VHH antibodies.**

A, B  Neutralization of SARS-CoV-2 B.1.351/Beta by the indicated VHH antibody constructs. The neutralization experiment was performed as described in Fig 3A, with the difference being that a mutant-optimized anti-RBD/S1 VHH cocktail was used for immunofluorescence staining (see Appendix Fig S13). Two titrations for the Re9F06 trimer are shown, corresponding to the standard (upper row) or a fivefold higher virus inoculum (lower row) that was used for even more stringent neutralization conditions.

**A**       **48h infection of Vero E6 cells with SARS CoV-2 B.1.351 (Beta / "South African")**

DNA
Spike protein (anti-RBD VHHs)
Spike protein (anti-S1ΔRBD VHHs)

**B**

Figure 10.

# Materials and Methods

### Alpaca immunization, generation of the nanobody library and phage display selection

Three female alpacas, held at the Max Planck Institute for Biophysical Chemistry, were immunized subcutaneously five times in weekly intervals with 33 μg of the entire SARS-CoV-2 S1 domain (C-terminally His-tagged) and 100 μg of the Spike receptor-binding domain (RBD, C-terminally fused to the Fc region of human IgG1), both obtained from Sino Biological (sinobiological.com, 40591-V08H and 40592-V02H, respectively). The generation of the nanobody library and phage display selection were performed essentially as described previously (Pleiner *et al*, 2015), using biotinylated RBD or S1 domain as baits, initially at 200 pM concentration, and 50 mM Tris/HCl pH 7.5, 500 mM NaCl, 1% (w/v) BSA as a selection buffer. The Re5, 6, 7, 9, 10, and 11 VHH series were selected against the RBD. The Re8 series was selected against S1 by pre-incubating phages with a 100-fold molar excess of unlabeled RBD. The Re5 and Re6 VHH series were obtained after two rounds of phage display, the Re7 series after three, and the Re8-Re11 series after four passes. For the Re9-Re11 series, the bait concentration was lowered to 30 pM. Recovered clones were identified by sequencing, and representatives of all classes were expressed as described below. In off-rate selections, 1 μM unlabeled Re5D06 was added 20 min after mixing phages with bait, followed by retrieving phages after various time points and evaluating the effects by sequencing individual clones.

### Cytoplasmic bacterial expression and purification of VHH antibody monomers and multimers

VHH antibodies that harbor just a single disulfide bond were produced as $His_{14}$-ScSUMO fusions by cytoplasmic expression in *E. coli* SHuffle® Express (New England Biolabs), which allows for formation of disulfide bonds in the cytoplasm. Cells were grown in 300 ml Terrific Broth (TB), and protein expression was induced at 21°C with 0.08 mM IPTG. Five hours after induction, 5 mM EDTA was added to the culture medium. The bacteria were then pelleted, resuspended in 50 mM Tris/HCl pH 7.5, 20 mM imidazole, 300 mM NaCl (lysis buffer) and frozen in liquid nitrogen. Cells were lysed by thawing and sonication, and insoluble material was removed by ultracentrifugation at ~ 160,000 *g* (~ 1 h, T647.5 rotor, Thermo Fisher Scientific).

Soluble material was applied to a 1-ml $Ni^{2+}$ chelate column. The matrix was washed with lysis buffer, with additional washing steps including 0.2% (w/v) Triton X-100 or 1 M NaCl, respectively. The VHH antibody was eluted by cleaving the $His_{14}$-SUMO-tag using 100 nM *S. cerevisiae* Ulp1p for 2 h at room temperature. The eluted VHH antibodies were frozen in liquid nitrogen and stored at −80°C until further use.

### Periplasmic bacterial expression and purification of VHH antibody monomers

VHH antibodies that contain two disulfide bonds (Re5E03, Re5E11, Re5G05, Re6E11, Re6F06, Re6G03, Re6H06, Re6H10, Re9F11) were expressed with an N-terminal pelB signal sequence and a C-terminal

$His_{10}$-tag in *E. coli* NEBExpress® (New England Biolabs). Cells were grown in TB medium (at 37°C); protein expression was induced with 0.05 mM IPTG for 2 h. Bacteria were harvested by centrifugation and lysed by osmotic shock lysis (Mergulhão *et al*, 2005). Periplasmic extract was recovered by two consecutive centrifugation steps at 4°C, first at 4,000 *g* for 20 min (F13 rotor, Thermo Fisher Scientific) and then at ~ 160,000 *g* for ~ 1 h (T647.5 rotor, Thermo Fisher Scientific). VHH antibodies were purified essentially as described above, but eluted with 50 mM Tris/HCl pH 7.5, 300 mM NaCl, 500 mM imidazole. Buffer was then exchanged to 50 mM Tris/HCl pH 7.5, 300 mM NaCl, 250 mM sucrose via a PD 10 desalting column (GE Healthcare). Aliquots were frozen in liquid nitrogen and stored at −80°C.

### VHH antibody production in yeast (*Pichia pastoris*)

A *P. pastoris* strain for high-yield secretion of C-terminally $His_8$-tagged Re6H06 into the culture medium was constructed essentially as described (Wu & Letchworth, 2004). The protein was expressed for 48 h by methanol induction at 30°C in methanol-complex medium (BMMY). Cells were pelleted by centrifugation, and the culture supernatant was then buffered with Tris/HCl pH 8.5. Re6H06-$His_8$ was purified by $Ni^{2+}$-chelate affinity and size exclusion chromatography (HiLoad 16/600 Superdex 75, Cytiva, equilibrated to 20 mM Tris/HCl pH 7.4, 150 mM NaCl), frozen in liquid nitrogen, and stored at −80°C until further use.

Re5D06 used for cryo-electron microscopy (Fig 6C) and for crystallization of free Re5D06 (Fig 6F) was also expressed in *P. pastoris*, but in an untagged form. Here, purification was done by affinity chromatography on a 5-ml Protein A column (HiTrap™ Protein A HP, Cytiva), equilibrated with 50 mM Tris/HCl pH 7.5, 300 mM NaCl. Following the binding step and one wash with equilibration buffer, the protein was eluted under acidic conditions (pH 2.05), neutralized with potassium phosphate, and applied to a gel filtration column (Superdex 75 HR 10/30, Pharmacia, equilibrated to 20 mM Tris/HCl, pH 7.2, 150 mM NaCl, 2% (w/v) glycerol). For crystallization, Re5D06 was concentrated to 33.3 mg/ml (2.32 mM). Aliquots were frozen in liquid nitrogen and stored at −80°C.

### Labeling of nanobodies

Nanobody monomers and trimers were labeled via two engineered cysteines (N- and C-terminal) per VHH using maleimides of Alexa Fluor 488, Alexa Fluor 568 or Atto 565 for immunofluorescence and $PEG_{11}$-Biotin maleimide (PEG1595; Iris Biotech) for BLI experiments, essentially as described previously (Pleiner *et al*, 2015).

### SARS-CoV-2 neutralization assay

For naming SARS-CoV-2 variants, we use the simplified nomenclature proposed by the World Health Organization (WHO), but also include the PANGO lineage identifier and the obsolete naming of variants by their geographic origin for completeness, since all of these naming conventions have been used in the literature cited in this work.

Vero E6 cells (Vero C1008) were maintained in Dulbecco's modified Eagle's medium GlutaMax™ (Gibco) supplemented with 10%

fetal bovine serum (FBS, Merck), 50 units/ml penicillin, 50 μg/ml streptomycin (Gibco), 10 μg/ml ciprofloxacin (Bayer), and 2 μg/ml tetracycline (Sigma). Cells were seeded into 96-well plates one day before SARS-CoV-2 infection/neutralization and cultivated at 37°C in a humidified atmosphere with 5% $CO_2$.

Nanobodies were serial-diluted on a separate 96-well plate, using supplemented DMEM GlutaMax™ medium (2% FBS). A patient-derived SARS-CoV-2 strain (D614G), which was previously sequenced and described (Stegmann *et al*, 2021) was used for virus infection. Virus stocks corresponding to $2.5 \times 10^6$ RNA copies of SARS-CoV-2 were pre-incubated with VHH antibodies 1 h prior to infection of cells. Medium, containing the nanobody-virus-mix was added to the cells and incubated for 48 hours at 37°C. SARS-CoV-2 B.1.351 (Beta variant) was obtained from the Robert Koch Institute (Germany) and neutralization experiments (Fig 10) were performed as described for "wild-type" SARS-CoV-2.

The cytopathic effect (CPE) caused by the virus was assessed by bright-field microscopy. For RNA isolation and virus inactivation, the SARS-CoV-2-containing cell culture supernatant was mixed with the Lysis Binding Buffer from the Magnapure LC Kit (# 03038505001; Roche, 1:1 ratio) and stored at −20°C. Cells were washed with PBS and fixed with 4% (w/v) formaldehyde in PBS for 1 h at room temperature, followed by a final wash with PBS and immunofluorescence staining (see below). All work with infectious SARS-CoV-2 was performed with permission by the responsible authorities after formal risk assessment and in a laboratory approved for biosafety level 3 work.

### Quantitative RT–PCR for virus quantification

The viral RNA was isolated using TRIzol™ LS (Thermo Fisher Scientific). After adding chloroform to the inactivated cell culture supernatant, the RNA-containing aqueous phase was isolated, and RNA was precipitated using isopropanol. The RNA pellet was washed with ethanol, dried and resuspended in nuclease-free water. Quantitative RT–PCR was performed according to a previously established assay involving a TaqMan probe (Corman *et al*, 2020), to quantify virus yield. The following primers were purchased from Eurofins:

| Primer | Sequence | Modification |
|---|---|---|
| P | ACA CTA GCC ATC CTT ACT GCG CTT CG | 5'FAM, 3'BBQ |
| F | ACA GGT ACG TTA ATA GTT AAT AGC GT | |
| R | ATA TTG CAG CAG TAC GCA CAC A | |

The amount of SARS-CoV-2 RNA found upon infection in the absence of VHH was defined as 100%, and the other RNA quantities were normalized accordingly.

### Immunofluorescence

Cells fixed after infection with SARS-CoV-2 were permeabilized for 3 min with PBS/0.15% (w/v) Triton X-100. After three washes with PBS, cells were blocked with 5% (w/v) BSA in PBS for ~ 1 h and washed once with PBS/1% (w/v) BSA. Staining was done with fluorescently labeled nanobodies diluted in PBS/1% (w/v) BSA for 1 h. For neutralization assays with "wild-type" SARS-

CoV-2, a mix of 15 nM Re8H11-Atto 565, 10 nM Re11C10-Alexa Fluor 488 and 15 nM Re11F11-Alexa Fluor 488 was used. The nanobody cocktail applied for immunostaining of cell infected with SARS-CoV-2 B.1.351 (Beta variant) included 5 nM of Alexa Fluor 488-labeld Re7E02, Re9C07, Re9G12, and Re9H03 and 15 nM Re8H11-Atto 565. Single nanobodies were used at 25 nM (Alexa Fluor 488) or 20 nM (Alexa Fluor 568) concentration (Fig 1B and C). After two washes with PBS, 2 μg/ml DAPI (in PBS) was applied for 5 min, followed by a final wash in PBS. Samples were immediately imaged on a Leica TCS SP8 confocal microscope (for 8-well slides) or a spinning disk confocal microscope (Visitron system based on Nikon optics) for neutralization experiments performed in 96-well format.

For epitope binning on transiently transfected HeLa cells (Fig 5A), permeabilization was performed with PBS/0.5% (w/v) saponin, but apart from this, the staining procedure was identical to the one described above. 5 nM Re10B10-Alexa Fluor 488 and 15 nM Re7E02-Alexa Fluor 568 were used for staining. Unlabeled competitors (150 nM VHH in PBS/1% (w/v) BSA) or just PBS/1% (w/v) BSA were applied to the permeabilized/BSA-blocked cells 20 min ahead of the staining mix. For VHH antibodies that are weak monomeric competitors, we used trimeric collagen XVIII NC1 fusions for competition (Re5A08, Re9F06, Re6B06, Re6D06). 96-well plates were imaged with a Zeiss LSM780 microscope (40x objective). Image analysis was done using Fiji (Schindelin *et al*, 2012).

### Expression and purification of the ternary Re9F06·RBD·Re5D06 complexes

The SARS-CoV-2 receptor-binding domain (RBD, residues 330-527 for the initial crystallization trials; 333-527 for the optimized, best-diffracting crystals) was co-expressed with VHH Re5D06 (from a bicistronic plasmid; TMP[R]) and VHH Re9F06 (Kan[R]) in *E. coli* SHuffle® Express (New England Biolabs) grown in TB medium at 25°C by induction with 0.15 mM IPTG. 3.5 h after induction, 5 mM EDTA was added to the culture medium, and the bacteria were pelleted, resuspended in 50 mM Tris/HCl pH 7.5, 300 mM NaCl, 20 mM imidazole (lysis buffer), frozen in liquid nitrogen, and lysed by thawing plus sonication. The cell lysate was cleared by ultracentrifugation at ~ 160,000 $g$ for ~ 1 h (T647.5 rotor, Thermo Fisher Scientific). The complex was purified by $Ni^{2+}$-chelate affinity chromatography, with elution by Ulp1p-mediated cleavage of the His[14]-ScSUMO tag on Re5D06. The eluted complex was passed over a gel filtration column (Superdex 75 HR 10/30, Pharmacia) equilibrated with 20 mM Tris/HCl pH 7.2, 150 mM NaCl (Appendix Fig S4A). For the optimized crystals, we used 20 mM Tris/HCl pH 7.0, 150 mM NaCl, 2% (w/v) glycerol. Peak eluate fractions were pooled and concentrated to 10–12 mg/ml for crystallization experiments.

### Limited proteolysis screening

All proteases (Trypsin, GluC/V8, Thermolysin, Chymotrypsin, Elastase) were purchased from Hampton Research. Proteolysis screening was performed with freshly prepared Re9F06·RBD·Re5D06 complex in 20 mM Tris/HCl pH 7.5, 150 mM NaCl. See Appendix Fig S4 for more details.

### Crystallization of Re9F06·RBD·Re5D06 complexes

All complex preparations (at 10–12 mg/ml protein) were screened by hanging-drop vapor diffusion (100 nl protein + 100 nl reservoir). An initial crystal hit was obtained with the Thermolysin-spiked Re9F06·RBD·Re5D06 complex (1:500 w/w, *in situ* proteolysis) in 100 mM MES, pH 6.5, 1.8 M ammonium sulfate, 10 mM $CoCl_2$. Crystals grown in larger drops (1.5 μl protein + 1.0 μl reservoir, Appendix Fig S4D) were cryo-protected with 20% (w/v) glycerol (added to the mother liquor) and plunge-frozen in liquid nitrogen. These crystals diffracted relatively poorly (~ 3.3 Å). Through construct optimization (see above) and additive screening, we obtained well-diffracting crystals with 0.1 M MOPS pH 7.5, 2.07 M ammonium sulfate, 0.1 M NDSB-256. These crystals were cryo-protected with 20% (w/v) glycerol.

### Crystallization of free Re5D06

Re5D06 purified from *P. pastoris* (33.3 mg/ml (2.32 mM) in 20 mM Tris/HCl, pH 7.2, 150 mM NaCl, 2% (w/v) glycerol) crystallized under a wide range of conditions. Hanging drops (100 nl protein + 100 nl reservoir) with 100 mM MES pH 6.0, 200 mM zinc acetate, 15% (v/v) ethanol yielded the best-diffracting crystals (see below). These crystals were cryo-protected in the precipitant solution.

### Structure determination

All diffraction data were collected at beamlines PXII-X10SA (Re9F06·RBD·Re5D06 complexes) and PXI-X06SA (free Re5D06) at the Swiss Light Source (SLS, Paul-Scherrer-Institut, Villigen, Switzerland) at 100 K. The Phenix Package (Liebschner *et al*, 2019) was used for structure solving.

For the Re9F06·RBD·Re5D06 complex that crystallized under *in situ* proteolysis conditions with Thermolysin (see above), we determined a low-resolution (3.3 Å) structure that was later used for phasing of high-resolution data of the optimized Re9F06·RBD·Re5D06 complex (see below). Three diffraction datasets from two crystals were merged, integrated, and scaled with the XDS package (Kabsch, 2010). These data showed significant anisotropy and were truncated using autoPROC (Vonrhein *et al*, 2011). The structure was solved by molecular replacement (MR) in PHASER (McCoy, 2007) using the published RBD and nanobody (H11-D4) models (PDB ID 6YZ5; Huo *et al*, 2020). For the VHH antibody search model, CDR loops were removed. Iterative cycles of manual model building and refinement were performed in Coot (Emsley *et al*, 2010) and in Phenix.refine (Afonine *et al*, 2012), respectively.

The high-resolution Re9F06·RBD·Re5D06 dataset was processed in XDS (Kabsch, 2010) and truncated using autoPROC (Vonrhein *et al*, 2011). The structure was solved by MR in Phaser (McCoy, 2007) with the lower-resolution structure of the complex as a search model. The model was manually adjusted in Coot (Emsley *et al*, 2010) and then subjected to refinement in Phenix.refine (Afonine *et al*, 2012).

The dataset for free Re5D06 was processed in XDS (Kabsch, 2010). The structure was solved by MR in Phaser (McCoy, 2007). A search model (with the CDR loops deleted) was created from the high-resolution structure of Re5D06 in the Re9F06·RBD·Re5D06 complex. Model building was performed with the Phenix AutoBuild

Wizard (Terwilliger *et al*, 2008) and in Coot (Emsley *et al*, 2010), followed by refinement in Phenix.refine (Afonine *et al*, 2012).

The quality of all final models was assessed with MolProbity (Williams *et al*, 2018). The structure was analyzed with PyMOL (The PyMOL Molecular Graphics System v.2.1, Schrödinger, LLC) and UCSF Chimera (Pettersen *et al*, 2004). Figures were prepared using UCSF Chimera or Coot.

### Expression and purification of SARS-CoV-2 Spike and free RBD

For Spike expression, we used the SARS-CoV-2 S HexaPro construct (Hsieh *et al*, 2020; Addgene #154754), which comprises the Spike ectodomain (1–1,208) with six exchanges to proline (F817P, A892P, A899P, A942P, K986P, V986P), a GSAS substitution at the furin cleavage site (residues 682–685), a C-terminal T4 foldon trimerization domain followed by a HRV 3C protease site, a $His_8$-tag and a Twin-Strep tag.

Expi293F cells ($10^6$ cells/ml, Thermo Fisher Scientific) were transiently transfected using Expifectamine and cultured for 6 days in the presence of 5 μM Kifunensine (Toronto Research Chemicals). Cultures were harvested by centrifugation. Supernatants were passed through a 0.22-μm filter, supplemented with buffer (1x PBS pH 7.4, 5 mM imidazole), and applied to $Ni^{2+}$-chelate beads. Beads were washed with 50 mM Tris/HCl pH 8.0, 150 mM NaCl, 30 mM imidazole, and bound protein was eluted with buffer containing 500 mM imidazole. The eluate was then incubated with Strep-Tactin Sepharose resin (IBA), washed with 50 mM Tris/HCl pH 8.0, 150 mM NaCl, and eluted with buffer supplemented with 50 mM Biotin. The protein was further purified on a Superose 6 10/300 column (GE Healthcare) equilibrated with 20 mM Tris/HCl pH 8.0, 150 mM NaCl.

For (biotinylated) RBD (wt and mutant) expressions in mammalian cells, we cloned the SARS-CoV-2 Spike RBD (residues 334–527) into the pHL-avitag3 vector (Aricescu *et al*, 2006; Addgene, 99847), which provides a C-terminal Avi-$His_6$ tag. Expression and purification were performed as described for the HexaPro Spike. For biotinylation, cells were co-transfected with a BirA expression vector and cultured in the presence of 100 μg/ml D-biotin (Roth). $Ni^{2+}$-chelate chromatography eluates were used in the BLI experiments without further purifications.

### Sample preparation for cryo-EM, grid freezing, and screening

Purified Spike and VHH Re5D06 (produced in *P. pastoris*, see above) were mixed in a 1:9 molar ratio (RBD-to-nanobody ratio of 1:3) and subjected to size exclusion chromatography (Superose 6 Increase 3.2/300, Cytiva; equilibrated with 2 mM Tris/HCl pH 8.0, 150 mM NaCl). 3 μl of the Spike·Re5D06 peak fraction (1 mg/ml; see Appendix Fig S8A) was applied to freshly (air) glow-discharged Quantifoil UltrAuFoil R2/2 grids (200 mesh). The grids were blotted at 4°C and 100% humidity for 3–6 s with a blot force of 5 and plunge-frozen in liquid ethane using a Vitrobot Mark IV (FEI Company). Grids were screened on a Glacios 200 kV TEM (Thermo Fisher Scientific) equipped with a Falcon III detector (Thermo Fisher Scientific). A blotting time of 4 s was optimal in this setup and yielded a gradient of ice thickness across the holes (Appendix Fig S8B), which was important for reaching good angular particle distribution coverage in the 3D reconstructions (Appendix Fig S8C and D, S10 and S11).

## Cryo-EM data collection, processing, and docking of the crystal structure

Cryo-EM data were collected with SerialEM (Mastronarde, 2005) using a Titan Krios G2 TEM (Thermo Fisher Scientific) operated at 300 kV, equipped with a Gatan GIF Quantum LS energy filter (with the slit width set to 20 eV) and a Gatan K3 Summit detector (operated in counting mode, non-super-resolution) at a nominal magnification of 81,000×, corresponding to a calibrated pixel size of 1.05 Å/pixel. Exposures were recorded over a defocus range set from −0.5 to −2.5 μm for 2.0 s with a dose rate of 22.00 e⁻·pixel⁻¹·s⁻¹, giving rise to a total dose of 39.91 e⁻/Å², which was fractionated over 40 movie frames. We recorded 15,636 movies by acquiring four movies per hole using beam-image-shift (with the data split in two batches of 4,255 and 11,381 movies, respectively).

Motion correction, dose weighting, CTF estimation and particle selection were done on-the-fly with Warp 1.0.9 (Tegunov & Cramer, 2019). Particles from batch 1 (~ 1.3 M) were extracted with Warp (box size: 420 pixels) and subjected to 2D classification in cryoSPARC 2.15 (Punjani *et al*, 2017). *Ab-initio* 3D models were generated from particles of good and bad 2D classes, respectively. Good particles were subjected to an initial 3D refinement, using the good *ab-initio* model (low pass-filtered to 60 Å) as a reference and further classified against good and bad *ab-initio* 3D models, yielding 990,627 good particles. Further 3D classification of these particles yielded 3 classes, two of which showed intact Spike·VHH complexes with good VHH occupancy. Class 1 (~ 420,000 particles) showed 3 RBDs in the open/up conformation while class 3 (~ 350,000 particles) represented Spike molecules with 2 RBDs in the open/up conformation and 1 RBD in the closed/down form. An initial 3D refinement was performed with both of these classes in cryoSPARC. Particles from batch 2 of the dataset (~ 3.4 M) were cleaned via the same classification pipeline (without the final 3D classification and refinement steps), yielding 2,472,178 good particles.

Good particles from both movie batches (3,462,805 total) were extracted in RELION 3.1 (Zivanov *et al*, 2018) at 4.2 Å/pixel (4x binned) with a box size of 106 pixels and subjected to 3D classification (Appendix Fig S9A). Two intact 3D classes emerged, both of which showed good VHH occupancy at the RBD: class 2 displayed two RBDs in an open/up and one RBD in a closed/down conformation, with the "down" RBD slightly rotated out relative to the "down" RBD in the Spike protein of PDB ID 7KMZ / EMD-22932 (Zhou *et al*, 2020) and the directly adjacent RBD tilted further outward. Class 5 displayed all RBDs in an open/up form. For subsequent 3D classification and refinement steps, unbinned particles of these two classes (box size of 424 pixels at 1.05 Å/pixel) were re-extracted in RELION, which was used for the remaining 3D classification and refinement steps that gave rise to maps 1; 2; and 3. The workflow is summarized in Appendix Fig S9. The maps were postprocessed in RELION using tight masks around the obtained volumes. The map sharpening B factors used are shown in Appendix Table S2. Local resolution plots, FSC curves, and angular distribution plots for the maps are shown in Appendix Figs S10A, S10B, and S11C, respectively.

The crystallographic model of the RBD·Re5D06 complex was rigid body-docked into map 3 in UCSF Chimera (Pettersen *et al*, 2004), rigid body-refined in PHENIX (Afonine *et al*, 2012), followed by minor (side chain rotamer) adjustments in Coot (Emsley *et al*, 2010). The cryo-EM map agrees well with the crystallographic model (Appendix Fig S11A and B).

## Kinetic measurements by Bio-layer interferometry (BLI)

BLI experiments were performed using High Precision Streptavidin biosensors and an Octet RED96e instrument (ForteBio/Sartorius) at 25°C with phosphate-buffered saline (PBS) pH 7.4, 0.02 % (w/v) Tween-20 and 0.1 % (w/v) BSA as assay buffer. Biotinylated proteins were immobilized on biosensors until a wavelength shift/binding signal of 1 nm (for the RBD) or 0.4 nm (for VHH monomers) or 0.75 nm (for VHH tandems) was reached. Subsequently, the biosensors were dipped into wells containing ligands for association and then incubated with assay buffer for dissociation. Data were reference-subtracted, and curves were fitted with a 1:1 binding, 2:1 binding or mass transport model (Octet Data Analysis HT 12.0 software). For low-affinity measurements, biotinylated RBD (see above) was immobilized on the sensors, dipped into wells containing 60, 30, 15, 7.5, 3.75, or 1.875 nM VHH for 300 s, followed by buffer for 600 s. For high-affinity measurements, biotinylated nanobodies were immobilized, incubated with 20, 10, 5, 2.5, 1.25 nM RBD (Z03479, GenScript) for 600 s, and followed by dissociation with buffer for 1 h. For RBD variant experiments, biotinylated nanobodies were immobilized, incubated with 20, 6.66, and 2.22 nM RBD wt (Z03479, GenScript) and mutants (Sino Biological; B.1.1.7/Alpha/ "UK" 40592-V08H82, B.1.351/Beta/"South African" 40592-V085H85, P.1/Gamma/"Brazilian" 40592-V08H86, and B.1.427/ B.1.429/Epsilon/Californian 40592-V08H28) for 450 s and then with buffer for 900 s. For testing the quadruple mutant (a combination of the Gamma/Brazilian and Epsilon/Californian variants), biotinylated RBD K417T, L452R, E484K, N501Y was immobilized and incubated with 100 nM nanobodies for 450 s, followed by buffer for 900 s. For thermostability BLI experiments, VHH antibodies (1 μM) were incubated at room temperature (RT) or at 90°C for 5 min and centrifuged for 20 min at 20,000 g. The supernatants were recovered and diluted 50-fold in running buffer (corresponding to 20 nM of the starting material). RBD-primed biosensors were dipped into wells containing VHHs for 450 s and then buffer for 900 s.

## ACE2 competition experiments

Biotinylated Human ACE2 (10108-H27B-B, Sino Biological) was immobilized on biosensors until a 0.5 nm wavelength shift/binding signal was reached. Biosensors were dipped into wells containing 50 nM RBD and 500 nM VHHs for 300 s.

## Size exclusion chromatography—multi-angle light scattering (SEC-MALS)

We used a setup in which a Superdex 200 10/30 GL column (GE Healthcare) is coupled to a UV sensor (1260 Infinity, Agilent Technologies, USA), a refractive index detector (Shodex RI-101, Showa Denko KK), and a miniDAWN TREOS static light scattering detector (Wyatt Technology). The Astra 6 software (Wyatt Technology) was used to calculate absolute molecular weights. The system was equilibrated with 20 mM Tris/HCl pH 7.5, 150 mM NaCl. The applied flow rate was 0.5 ml/min.

**Differential scanning fluorometry (DSF/Thermofluor)**

Nanobodies were diluted to 1 mg/ml in 20 μl 50 mM Tris/HCl pH 8.0, 300 mM NaCl, 1x SYPRO Orange (Life Technologies). Samples were pipetted into a Hard-Shell® 96-well plate (Bio-Rad). The plate was sealed with transparent MicroSeal® "B" Seal (Bio-Rad) and briefly centrifuged to remove air bubbles. Experiments were performed using the CFX96 Real-Time System (C1000 Thermal Cycler, Bio-Rad). The samples were incubated for 5 min at 20°C before the temperature was gradually increased to 95°C with 1-K increments and 45 s for each incubation step. At the end of each step, the SYPRO Orange fluorescence was measured using the HEX channel. Melting temperatures are defined as the inflection point of the melting curve before reaching the first melting peak. VHH variants were considered hyperthermostable if they produced no melting peak.

**Software used for preparing figures**

All structural representations have been generated with UCSF Chimera (Pettersen *et al*, 2004) or Coot (Emsley *et al*, 2010). Figures were prepared using Photoshop and Illustrator (Adobe Systems Inc.). Graphs were prepared with Microsoft Excel and GraphPad Prism (version 9.1.1 for macOS, GraphPad Software, San Diego, California USA, www.graphpad.com). See above for additional software used.

# Data availability

Coordinates and structure factors have been deposited with the Protein Data Bank (PDB, https://www.rcsb.org/) with accession codes 7OLZ (Re9F06·RBD·Re5D06 complex) and 7ON5 (free Re5D06). The cryo-EM reconstructions were deposited with the Electron Microscopy Data Bank (EMDB, https://emdb-empiar.org/) under accession codes EMD-13107 (SARS-CoV-2 Spike·Re5D06 complex in an "all-RBDs-up" conformation; map 1), EMD-13106 (SARS-CoV-2 Spike·Re5D06 complex in a "one-RBD-down" conformation; map 2) and EMD-13105 (SARS-CoV-2 RBD·Re5D06 subvolume from the "one-RBD-down" conformation of the Spike; map 3). Sequences of nanobody constructs are available upon request.

**Expanded View** for this article is available online.

## Acknowledgements

We thank Leonie Neumann and Svetlana Agafonova for excellent technical help, Antonio Politi (imaging facility of the MPI-BPC) for setting up an automated imaging workflow for neutralization assays in 96-well format, Ulrich Steuerwald and Jürgen Wawrzinek (crystallization facility of the MPI-BPC), Christian Dienemann and Ulrich Steuerwald for maintenance of the cryo-EM facility at the MPI-BPC, Ralf Rümenapf and the team of the MPI-BPC animal facility for alpaca care, Sarah Kimmina for veterinary support, Vasundara Srinivasan for help with freezing the initial crystals of the Re9F06·RBD·Re5D06 complex, the team of the SLS beamline (in particular Vincent Olieric) for superb support during x-ray data collection, and Uwe Pleßmann and Henning Urlaub for mass spectrometry, as well as Thorsten Wolff (Robert Koch Institute, Berlin) for sharing the B.1.351 virus strain. We thank Johannes Bange (Lead Discovery Center, Dortmund), Dieter Link (Max Planck Innovation), and Holm Keller (kENUP Foundation) for helpful discussions, as well as the Max Planck Society and the Max Planck Foundation for generous financial support. Open Access funding enabled and organized by Projekt DEAL.

## Author contributions

Conception and design of the study/experiments: DG, MD, TG, MA, AD, KMS, OR, and VCC; vaccine design/preparation: DG, PG, TG, BM, and JK; alpaca immunizations and blood sampling: UT, BM, and JK; veterinary care: UT; library generation and phage display: RR, BM; production of nanobody constructs: KG, RR, WT, JS, OR, TG, and DG; biophysical experiments (BLI, DSF, and SEC-MALS): MA, JS, KG, and TG; neutralization experiments: AD, KMS, MD, TG, DG, KG, PG, VCC, and UG; immunofluorescence: KG; crystallization: TG, WT; X-ray data acquisition and structure determination: MA and TG; mammalian expression/purification: MA; cryo-EM work: TG, MA, and CD; data analysis and interpretation/figure preparation: DG, TG, MA, AD, KMS, and MD; drafting and revising the manuscript: DG, MD, TG, MA, and VCC. All authors read and commented on the paper.

## Conflict of interest

TG, MA, AD, KMS, KG, RR; WT, OR, JS, PG, BM, JK, UT, VCC, MD, and DG are inventors on a patent application encompassing the anti-Spike nanobodies described in this study. TG, MA, RR, KG, WT, OR, and DG are inventors on a patent application on fold-promoting nanobodies.

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
