## [Review Process File · The EMBO Journal]

Neutralization of SARS-CoV-2 by highly potent, hyper-thermostable, and mutation-tolerant nanobodies

Thomas Güttler, Metin Aksu, Antje Dickmanns, Kim Stegmann, Kathrin Gregor, Renate Rees, Waltraud Taxer, Oleh Rymarenko, Jürgen Schönemann, Christian Dienemann, Philip Gunkel, Bianka Mussil, Jens Krull, Ulrike Teichmann, Uwe Groß, Volker Cordes, Matthias Dobbelstein, and Dirk Görlich

DOI: 10.15252/embj.2021107985

Corresponding author(s): Dirk Görlich (goerlich@mpibpc.mpg.de) , Matthias Dobbelstein (mdobbel@uni-goettingen.de)

Review Timeline:

Submission Date:	12th Feb 21
Editorial Decision:	16th Mar 21
Revision Received:	31st May 21
Editorial Decision:	17th Jun 21
Revision Received:	6th Jul 21
Accepted:	13th Jul 21

Editor: Karin Dumstrei

Transaction Report:

Dear Dirk,

Thank you for the constructive discussions yesterday regarding the revisions on the manuscript and what can be done to address the raised concerns.

I appreciate the outlined approach and would like to invite you to submit a revised version.

When preparing your letter of response to the referees' comments, please bear in mind that this will form part of the Review Process File, and will therefore be available online to the community. For more details on our Transparent Editorial Process, please visit our website:

<https://www.embopress.org/page/journal/14602075/authorguide#transparentprocess>

Let me know if we need to discuss anything further.

With best wishes

Karin

Karin Dumstrei, PhD
Senior Editor
The EMBO Journal

- a point-by-point response to the referees' comments, with a detailed description of the changes made (as a word file).

- a word file of the manuscript text.

- individual production quality figure files (one file per figure)

- a complete author checklist, which you can download from our author guidelines (<https://www.embopress.org/page/journal/14602075/authorguide>).

- Expanded View files (replacing Supplementary Information)

The revision must be submitted online within 90 days; please click on the link below to submit the revision online before 14th Jun 2021.

Referee #1:

In this manuscript, Guttler et al. describe the isolation and characterization of a large panel of anti-SARS-CoV-2 nanobodies, with a number of them displaying extremely high neutralizing potency, as well as a very high stability. The authors also use a trimerization approach that increases the potency of some of their nanobodies by several logs. The paper presents an impressive amount of data obtained using a variety of state-of-the-art procedures to characterize the nanobodies. It also describes very clever selection strategies for the identification of the most potent nanobodies as well as a very thorough biophysical, structural and biological characterization. The paper also reports a high level of protein engineering to optimize the nanobodies thermal stability and neutralization potency, resulting in a short list of leading candidates with properties amenable to clinical applications. Altogether, these highlights make the paper a strong candidate for publication in the EMBO Journal.

Overall, the article is well written, has a clear goal - which is highly relevant in the context of the current COVID-19 pandemic. Yet there are several aspects that the authors should address before accepting this manuscript:

Manuscript presentation:

- The merge of Results and Discussion does not work to the benefit of the article. The authors discuss in long paragraphs ideas that, in some cases, are debatable speculation (see below) and dilute the important message that can be drawn from the experiment.

Experimental data:

- In lines 108-109, it is reported that the binding kinetics of the identified VHHs were determined by BLI, but only a few are reported (In Figure 2). In the preceding paragraph, it is written that 58 nanobodies in total were identified. It is important to provide a supplementary Table with the BLI results listed for all the characterized nanobodies. Also, lines 169-175 report that all antibodies were tested for neutralization at 500nM, so at least the neutralization % at this concentration could be given for them. The supplementary table could provide the kinetic parameters (K_{on} , K_{off} , K_D , Neutralization IC_{50} (and IC_{99}), or the percentage of neutralization observed at 500nM, for each nanobody - or at least for all those for which it was determined. Also, the melting temperature (T_m) whenever available.

- The authors present a "preliminary" analysis of their cryo-EM data (limited to a supplementary

figure with a low-passed map) of the SARS-CoV-2 spike in complex with the same two nanobodies for which they determined the structure. It is important to finish the cryo-EM data processing and to present in the revised manuscript a final structure with proper statistics. The authors quote to have collected many images on a Titan microscope and to have over 4 million particles in total, so it should be straightforward to obtain a high-resolution model. These results will add a lot of value to the paper.

- The authors discuss the emergence of new SARS-CoV-2 variants and speculate how the mutations in the RBD would impact the binding to one nanobody, but they don't provide any experimental support on the efficacy of the antibodies in recognizing and neutralizing the variants. They insist throughout the manuscript on the characteristics that make their nanobodies so adequate for clinical purposes, but they leave out proving that they will work against the current variants that are circulating in many countries. It is therefore important to provide at least some BLI data showing that the VHHs, at least, bind to the mutant RBDs (even better if they provide neutralization results, just binding to an isolated RBD may not recapitulate the action on the full Spike and neutralization of the infection).

- Regarding the analysis that they perform on the potential effects of the SARS-CoV-2 variants, there are some aspects that were left out of the discussion:

o Lines 318-321: they don't expect changes in binding to the South African and Brazilian strains, because the aliphatic chain of the introduced K484 would make a favorable contact to Y109/W110 in the VHH. However, the E484 in the Wuhan strain forms hydrogen bonds with residues on the VHH and the authors don't discuss if these bonds can be formed with a lysine residue. Moreover, they completely omit discussing the effect of the K417N/T mutation, when K417 makes a direct salt bridge with E113 in the nanobody (Fig. S13). In sum, their discussion is incomplete and, possibly, too optimistic.

- The only insights on the nanobodies neutralization mechanism come from the crystal structure they report. It is quite clear that the antibodies targeting epitope 1 will block ACE2 binding, but this is not so obvious for those directed towards epitope 2 (the overlapped region with ACE2 is rather small). The manuscript would benefit from competition experiments (which can be performed by BLI, a technique that the authors use) to experimentally demonstrate that blocking ACE2 binding is the mechanism of action of the panel of nanobodies, with particular emphasis on those against epitope 2.

- There is a very brief structural description on the epitope recognized by Re6A11. This seems a missed opportunity, since the authors report that the entire panel is recognizing basically two regions, and the focus of the analysis is put only on epitope 1. I think the authors should include a structural description of epitope 2 beyond the identification of contacting residues (Fig. 6d), as was done for Re5D06. In this sense, the discussion would benefit from a comparison of both epitopes to those of previously reported nanobodies (is there any novel epitope, or everyone is isolating antibodies solely against two regions?) and mention if epitope 2 is conserved or not.

- The structures should be deposited in the PDB (and the map in the EMDB) before submission and provide the corresponding codes. Similarly, the amino acid sequences of the 58 characterized nanobodies should be provided.

- The paragraph in lines 410-418 makes several statements for which the data are not provided. There should be a table listing the data on the different constructs mentioned and their IC50, in particular to show the performance of the different collagen domains used and the corresponding linkers. Also, it is important to provide the amino acid sequences of the linkers used. Including all the data would better reflect the tremendous amount of protein engineering that was performed and would be very useful for the reader. In addition, the authors should comment on the expected stability of the linker constructs vis-à-vis potential proteolysis (i.e., how stable are they with respect to fortuitous cleavage by circulating proteases when injected into a living organism)

Minor comments:

- The names used for the nanobodies are very hard to follow, sometimes they differ in a single number or a letter. I suggest that the authors come up with a better scheme. They could, for instance, use the domain and the epitope for those targeting the RBD (R1_01, R1_02, etc. and R2_01, R2_02, etc. for those targeting epitopes 1 and 2, respectively. And N01, N02, etc., for those targeting epitopes outside the RBD. The order (01, 02, could be an ascending order based on the measured Kd. Such a scheme would make the paper much easier to follow than it is currently the case.
- The authors use the term "symmetry-matching" nanobodies (in the title and in several parts of the manuscript; lines 25, 76, 78, for example), but I am not sure this is accurate. The first time I read it I had the idea that it would refer to an object that matches exactly the position of the RBDs in the Spike (like a plug and a socket). However, the trimeric nanobodies present flexible linkers, so overall the molecule does not present a C3 symmetry. It would be more appropriate if the authors just called them "trimerized nanobodies", as they could also cross-link spikes at the virion surface, and bind in a non-symmetry matching pattern. If the authors wish to make a difference with respect to other strategies, for instance that VHHs linked via G4S linkers, they should state this difference and mention that their antibodies are trivalent and trimeric, while the ones produced with linkers are trivalent and monomeric.
- The authors propose their VHH+RBD co-expression strategy for a simplified production of vaccines and adaptation to viral escape-mutations (lines 29-30, 81-82, 263-277), which is debatable. First, if an escape mutant appears the "folding" antibody may not bind anymore and that would be the end of the production system. Second, the co-expression leads to co-purification, so the immunogen wouldn't be just the RBD, but the complex RBD+VHH.
- The term "force" is not well used (line 28, 79). It's better to use "promote", as was done in several sentences.
- The wording in lines 54-58, which compare vaccines and passive immunization, is confusing, as it appears to suggest that the administration of antibodies could be done in a scale capable to stop the spread of a virus. This paragraph should be revisited.
- Line 66: "combinatorial issues" probably refer to combination of light and heavy chains, but it is ambiguous, as there are also 'combinatorial issues' at the VDJ recombination level.
- Was the affinity of anti-S1, non-RBD nanobodies determined? If so, they should be included in the supplementary Table requested above. This should also be mentioned to rule out that all these nanobodies are non-neutralizing only because they don't bind with high affinity. Also, the terminology used to describe the methodology used to select them is confusing: S1ΔRBD suggests that the authors have made a construct of S1 in which the RBD was deleted, which is not the case. Calling these Mabs just non-RBD binders is enough, as S1 was the only immunogen used to obtain the nanobodies.
- Was the k_{on} and k_{off} measured by BLI or SPR? SPR is often a better method for obtaining kinetic parameters and it seems the authors used both, but the SPR was presented only on Fig. 7e.
- Lines 176-183: unnecessary, it's quite obvious that the antibodies should neutralize at the lowest dose possible.
- Lines 191-194: not necessary.
- Line 229: "trimeric complex". It is a ternary complex, not a trimer since the three objects in the complex are different.
- The Materials and Methods section does not contain cloning information, and this should be included. Were the RBD, Re6A11 and Re5D06 cloned in three plasmids? In one plasmid? Which was the plasmid? Which was the plasmid used to express the other nanobodies?
- Lines 249-262: I don't see a point on keeping this in the paper, but in case of doing so it should be on a separate Discussion section.
- Lines 263-277: completely speculative and should be removed from the manuscript. The fact that

co-expressing the VHH allowed the authors to obtain a complex and solve a structure is reason enough to include the strategy in the paper, there is no need to boost it with other debatable applications.

- Lines 285-286: "cation-pi interactions". According to Fig S13 these interactions are between residues in the VHH, not with the RBD.
- Lines 322-327 and Fig.6c: prepare the figure with the results of the experimental cryo-EM structure.
- Line 356: it is not clear to me if Re5D06R13 has all the mutations mentioned in the previous paragraph, plus T58V and V93D. Does it have, in total, 10 mutations? Maybe including the list of all the changed residues in the figure could help, or as part of Materials and Methods (this applies to Re5D06R15 too).
- Lines 372-374: the sentence regarding the potency should be toned down. The paper cited on reference 24 presents monomeric VHHs with IC50 of 22 and 48pM (not the same as IC99+, but it sets a "precedent" among monomeric VHHs).
- Line 381: proposes a second disulfide on Re6H06 as the source of high thermal stability. Is this observed for all the antibodies with a second disulfide bond (Re5E03, Re5E11, Re5G05, Re6E11, Re6F06, etc.)? If not, remove this statement or discuss reasons for such a discrepancy.
- Lines 385-388: probably match better the following paragraph.
- Line 403-404: "it should be small for the sake of material economy". I don't understand this point... it should be small to save amino acids in the bacteria?? It seems better to delete this phrase.
- Lines 423-425: it is proposed that trimerization, instead of a dimeric Fc, is the responsible for a large increase in VHH-72 potency. Is this high-magnitude increase also obtained if VHH-72 is trimerized with collagen XV? If so, this favors the hypothesis, but if it's not the case then the statement should be revised.
- Lines 429-434: it seems an explanation on avidity, which every reader already knows.
- Is the outstanding thermal stability maintained in the trimeric design?
- Overall, the "Outlook" section is too long, it should be changed into a Discussion and should be more focused.
- Line 537: "66 kDa is well above... 60 kDa". It doesn't seem so "well above" to me. It's more like "barely the size...".
- Lines 544-547: "... it is improbable that a given mutant can escape many independent binders at the same time". I agree with this statement, but in the context of this article the different binders don't seem to be so "independent" since the 60 VHHs recognize basically two epitopes. Therefore, some single mutations could affect multiple binders at the same time. These considerations should be included in the discussion.
- Line 562: among the examples of three-fold symmetry the authors mention Dengue and Zika viruses. They should be removed, since their envelope glycoproteins are displayed on the viral surface with different symmetry axes, they don't represent the trimeric arrangement seen on fusion proteins like those in Influenza, Coronaviruses, etc.
- Fig.6 and Fig. S13: it would be clearer if the dashed lines are used only to indicate polar contacts.
- Figure 1: it would benefit of an inset in the figure with the color coding (it's easier than reading the figure legend).
- Figure 8: include the theoretical molecular weight of each protein in their monomeric form (in this way, the reader can know that 60 kDa corresponds to a trimer).

Typos:

- Line 113: extra parenthesis (or one missing).
- Line 115: "released from in the dissociation" (just "released" or "released from it").
- Line 280: ACE should be ACE2

Referee #2:

Güttler et al report the properties of nanobodies raised in alpaca by immunization with SARS-CoV-2 S1 fragment and SARS-CoV-2 RBD. They have found several that bind the RBD quite tightly (single-digit nM) and that neutralize (in a cell assay) with corresponding potency (ca 5-10 mg/L). They also claim to have found even more tightly binding nanobodies, but see my concerns below. A structure of the RBD with two non-competing neutralizing nanobodies bound shows the tight fit and extensive solvent-excluding interfaces that give rise to high affinity. They have constructed a trimer of one of them, using a procollagen trimerization element, and as expected shown substantial increase in affinity and neutralization potency. Finally, because nanobodies can be expressed in good folded yield in E coli, but RBD ordinarily cannot, they find that at least one class of the nanobodies is an effective folding chaperone for the RBD.

The key problem with the paper is that it is all over the place, and flawed in each of those places -- probably not irreversibly, but enough to weaken substantially each part of it, leaving very little unflawed.

- (1) The affinity measurements are flawed, because (unless Fig. 2 is misleading) they seem to have used only one concentration. The apparent dissociation rate constant is indeed low, but probably precisely because of mass transport -- which they seem to misunderstand. Mass transport will give apparently slower dissociation rate constants, because of vicinal rebinding while the ligand still contributes to the refractive index difference being detected. So unless I misread Fig. 2, the only ones of those numbers I can believe are those with nM KD and reasonably measurable koff, although I'm sure that the trimer does bind more tightly than the corresponding monomer.
- (2) The structure analysis is fine as it stands, but only in the interest of further "drug design". We already know that there are several distinct immunogenic regions on the RBD. For example, the two Regeneron antibodies whose reputations were either enhanced or tarnished, depending on your point of view, by being touted by infamous members of the former US administration, do not compete, and the crystal structure of those two antibodies bound together on the the RBD was reported almost a year ago. See also Piccoli et al, Cell, Nov. 20, 2020, and Barnes et al, Nature, Oct. 12, 2020, for very thorough analyses of sites of the RBD.
- (3) The authors make a reasonable claim for their preferred neutralization assay, but they do not seem to have validated it carefully. Was the nanobody removed before incubating the cells for 48 hrs -- otherwise, it could have been preventing spread, depending on the initial moi.
- (4) There are a number of statements that appear to show naiveté about immunology and antibody binding. (a) Divalent antibodies bind tightly not only because sometimes they can bridge two sites, but also because they rebind rapidly, presumably due to the presence of the second site at high local concentration. Most of us who work with both Fabs and IgGs find that the IgG from which the Fab was derived binds with 100 to 1000-fold lower KD (more tightly), as measured by SPR or BLI, than does the Fab. (b) There is no evidence for SARS-CoV-2 that most of the spikes need to be antibody bound to neutralize. For those viruses (e.g., influenza) for which detailed studies have been done, one needs far less than full occupancy, depending on the epitope. (c) The evolution of vertebrate immune systems (non-agnathan) is based on dimeric antibodies, so suggesting that dimers are not "good enough" seems a bit curious. Of course, from the point of view of designing a therapeutic, the trimer strategy is sensible (and has already been adopted and published for ACE2 as an inhibitor -- Xiao et al, NSMB Feb. 2021 but already in bioRxiv Sep 2020). (d) The required serum concentration of a therapeutic for reasonable clearance will probably depend on serum lifetime, availability at the lung mucosa, etc., so the arguments about concentration, negligible elimination, etc., are oversimplified and not really relevant at this stage.

(5) Production of the RBD in E coli because of the chaperone effect of the nanobody is the most interesting, perhaps the only really new, contribution. The rationale is a bit flawed -- RNA vaccines are almost instantly adaptable, as all one does is change the DNA template sequence, with little further production optimization needed, and the adapted vaccine can probably be tested expeditiously, as new phase 1 trials would be minimal or unnecessary because nothing would be different except the sequence of the RNA (and the expressed immunogen in the subject, so the issue of autoimmunity would always be there, but not generally detectable in a limited phase 1 trial anyway). What this group should explore, given the finding and given who they are, is the general issue of whether co-expression with nanobodies can allow one to express single-domain eukaryotic proteins in E coli that cannot be obtained on their own in good yield. That would be a substantial advance and a lasting contribution. Moreover, the RBD alone, or even oligomerized on a scaffold, may not be an optimal immunogen. In the early, preclinical trials of the J&J vaccine, several different constructs were examined, including RBD, but intact spike was ultimately found to be preferable.

Referee #3:

Guttler and colleagues have developed a series of single chain Llama antibodies raised against SARS-CoV-2 spike with a view to using them as therapeutics in COVID19. Fig 1 shows a phylogenetic tree of the Ab. Fig 2 shows binding data for the best Ab. Fig 3 and 4 show nM neutralisation data by immunofluorescent staining and QPCR to detect infected cells. Fig 5 uses IF neutralisation data to illustrate that the Ab bind 2 distinct areas on spike allowing 2 competition groups to be described. Fig 6 shows structures of the key Ab in complex with the spike RBD and shows how neutralisation is achieved. Fig 7 examines thermostability and shows how this can be improved by mutagenesis of specific Ab. Fig 8 demonstrates how the Ab can be trimerized to improve neutralisation though improving affinity. The authors also show how co-expression of the Ab with Spike RBD improve RBD production and discuss how this may improve Antigen production for vaccines.

The work is exciting, well presented and the experiments are compelling and well controlled.

I have only 1 major point, point 1, followed by minor suggestions to improve clarity.

1. There is no discussion of repeatability, how many times the experiments were repeated and how reliable this is. I'm sure its good but this needs to be shown. How representative are the neutralisation assays shown. These are usually plotted with error bars but here we just see one IF example. The QPCR bars are also apparently only done once with no error bars. Can we be reassured that this has been done more than once and some kind of statistical support be provided.

2. The authors might want to discuss how their work contributes to future pandemic preparedness, ie having Ab against Coronaviruses ready to go therapeutically. Is there any comment they can make on broadness, ie their Ab hitting other coronas such as CoV1 or MERS?

3. Line 44. I think that poor activity of remdesivir and direct acting antivirals is due to the fact that by the time someone shows up in hospital the virus has done the damage. Its not because remdesivir acts later in the lifecycle. I ask the authors to reconsider this point.

4. Para beginning line 131. This text is confusing because it talks about using fixed virus but then

talks about infected cells, CPE and released virus. Please clarify how this was done.

5. The description of Fig 5 is hard to follow because there's no panels distinguished. Can you clarify this by giving the panels letters and referring to them individually. Generally what is competing with what could be more clearly described.

6. From line 251 to 277 could be moved to discussion to focus on the data. Certainly more separation of results and discussion and some shortening of the discussion where possible would improve clarity. For example, more succinctly summarise the general advantages of nanobodies rather than review the field, and then focus on discussion of the specific Ab described herein.

7. Outlook should be discussion I guess.

Typos/grammer

8. Line 34 the virus infected more (no had)

9. Line 55 However, to be effective....

10. Line 115 released in the (no from)

11. Line 538 cut off at

Major changes to the manuscript

- The new Fig 2 now shows RBD-affinities of eight nanobodies measured by kinetic bio-layer interferometry (BLI). This includes three main epitope binders with ~1 pM affinities. We also identified a high-affinity (~30 pM) binder (Re5F10) to the fold-promoter epitope.
- We tested by BLI which nanobodies block the ACE2-RBD interaction. The outcome was that all main epitope binders (e.g., Re5D06, Re6H06, Re9B09) and some epitope 2-binders (e.g., Re5F10 or Re9F06) do, while other epitope 2-binders (e.g., Re9C07 or Re9G12) do not. This data is shown in the new Figure 5B.
- We have expanded the neutralization dataset, now show duplicates (for Figures 3-4) in the Appendix (Fig S1) and include the number of replicates for all relevant neutralization experiments.
- We have expanded the description of the RBD-Re6A11 interaction (to simplify terminology, we now use the same term for Re6A11 and Re9F06 since the two differ only by an irrelevant Q-L exchange). Appendix Fig S5 now depicts the molecular details of the RBD-Re9F06 interaction.
- We have improved the cryo-EM reconstructions of the Re5D06:Spike complex classes from an initial resolution of 10 Å (as previously shown in the supplementary material) to an average resolution of 2.8 Å for each of the two classes represented in the dataset. We further refined the RBD-Re5D06 module of one of the classes to a higher local resolution (3.5 Å). All resulting maps agree with the crystallographic model. The cryo-EM data is shown in Fig 6C and Appendix Figures S8-S11.
- We have included more thermostability data (new Fig 7) and now show, for several nanobodies, that they tolerate heating to 90 °C without impairment of subsequent RBD binding in a BLI setup.
- We have included a comprehensive BLI dataset on nanobody binding to mutant RBDs (new Fig 9). These mutants include the UK B.1.1.7 strain, the South African B.1.351, the Brazilian P1 strain, the Californian B.1.429 strain, as well as a combination of the Brazilian and Californian mutations (K417T, E484K, N501Y, and L452R). We identified one nanobody monomer (Re9H03) that binds the quadruple mutant still with low picomolar affinity. Likewise, tandem fusions also tolerated this extreme mutant combination.
- We now include neutralization data for the South African B.1.351 strain and show ~50 picomolar neutralization for Re9H03, Re6H06, as well as for Re9F06-Re5D06 and Re9F06-Re9B09 tandems (new Fig 10). The so far best fold promoter (Re5F10) neutralizes this mutant at 1.7nM. A Re9F06 trimer neutralizes to 5-17 pM concentration. We observed the same neutralization potency for a Re9F06-Re6H06 tandem fusion.
- The text was re-arranged to accommodate these changes. All discussion of virus mutants is now based on actual data. Numerous smaller adaptations were implemented (also in response to the reviewers' comments). The previous Outlook section was condensed and merged with the Results and Discussion section.
- We included additional information in the supporting material file to address the questions raised by the reviewers (see our response to the reviewers' queries).
- We have included discussions of anti-RBD-nanobody structures previously published by other groups. There is, for example, a striking similarity between our main epitope binder Re5D06 and VHH E of König et al., with the difference that Re5D06 shows a 1000 times better affinity and a 1000 times higher neutralization potency.
- We also discuss the leads of the paper by Xiang et al. and their susceptibility to the E484K mutation.

Answers to the Reviewers' queries

(for clarity, we repeat the Referees' feedback in blue in front of each of our answers)

Referee #1

In this manuscript, Guttler et al. describe the isolation and characterization of a large panel of anti-SARS-CoV-2 nanobodies, with a number of them displaying extremely high neutralizing potency, as well as a very high stability. The authors also use a trimerization approach that increases the potency of some of their nanobodies by several logs. The paper presents an impressive amount of data obtained using a variety of state-of-the-art procedures to characterize the nanobodies. It also describes very clever selection strategies for the identification of the most potent nanobodies as well as a very thorough biophysical, structural and biological characterization. The paper also reports a high level of protein engineering to optimize the nanobodies thermal stability and neutralization potency, resulting in a short list of leading candidates with properties amenable to clinical applications. Altogether, these highlights make the paper a strong candidate for publication in the EMBO Journal.

We appreciate the accurate summary and the highly positive overall evaluation.

Overall, the article is well written, has a clear goal - which is highly relevant in the context of the current COVID-19 pandemic. Yet there are several aspects that the authors should address before accepting this manuscript:

Manuscript presentation:

- The merge of Results and Discussion does not work to the benefit of the article. The authors discuss in long paragraphs ideas that, in some cases, are debatable speculation (see below) and dilute the important message that can be drawn from the experiment.

We have re-organised the manuscript, in particular as we now have replaced speculations on the virus mutants by experimental data. These include:

- BLI data on nanobody binding to mutant RBDs (K417T/N, L452R, E484K, N501Y)
- Identification of nanobodies that are particularly resistant to the current escape mutations
- and even neutralization data of the South African B.1.135 mutant strain (see below).
- We have also condensed and re-written the outlook section and merged it with the Results & Discussion section. Having a separate Discussion, revisiting even just the most important findings, is not going to work because the manuscript is already extremely long (105 000 characters without supplements).

Experimental data:

- In lines 108-109, it is reported that the binding kinetics of the identified VHHs were determined by BLI, but only a few are reported (In Figure 2). In the preceding paragraph, it is written that 58 nanobodies in total were identified. It is important to provide a supplementary Table with the BLI results listed for all the characterized nanobodies. Also, lines 169-175 report that all antibodies were tested for neutralization at 500nM, so at least the neutralization % at this concentration could be given for them. The supplementary table could provide the kinetic parameters (Kon, Koff, KD, Neutralization IC50 (and IC99), or the percentage of neutralization observed at 500nM, for each nanobody - or at least for all those for which it was determined. Also, the melting temperature (Tm) whenever available.

We have now added a considerable body of BLI, thermostability and even neutralization data (see Figures 2A and B, parts of 4A-D, 5B, 7, 9, 10).

Whenever we talk about neutralization, we mean complete neutralization. The reason is that we employ a multi-round infection assay, which shows either full infection or a complete block infection. We very rarely see partial neutralization where just a few cells get infected.

We used initial rounds of characterization, such as the above-mentioned neutralization assay at 500nM nanobody concentration, to narrow down a list of interesting ones, which we then characterized in more detail. However, it was not possible within reason to characterize all the nanobodies and all nanobody derivatives in the same depth, since these are hundreds by now. It would therefore be rather misleading to list all of them in the same table. Instead, we only included two tables (Appendix Tables S3-4) that show selected nanobodies and their properties. In turn, we toned down the statement on our cursory initial characterization. In fact, this was only meant to be transparent about our decision process.

- The authors present a "preliminary" analysis of their cryo-EM data (limited to a supplementary figure with a low-passed map) of the SARS-CoV-2 spike in complex with the same two nanobodies for which they determined the structure. It is important to finish the cryo-EM data processing and to present in the revised manuscript a final structure with proper statistics. The authors quote to have collected many images on a Titan microscope and to have over 4 million particles in total, so it should be straightforward to obtain a high-resolution model. These result wil add a lot of value to the paper.

We have improved the cryo-EM reconstructions of the Re5D06-Spike complex classes from an initial resolution of 10 Å (as previously shown in the supplementary material) to an average resolution of 2.8Å for each of the two good classes represented in the dataset.

The RBD-Re5D06 module shows some flexibility in respect to the more rigid core structure of the Spike. This limits resolution. Nevertheless, we were able to refine the RBD-Re5D06 volume of one of the classes to a local resolution of 3.5 Å. All resulting maps are in agreement with the high-resolution crystallographic model. We present the cryo-EM data in Fig 6C and Appendix Figures S8-11.

- The authors discuss the emergence of new SARS-CoV-2 variants and speculate how the mutations in the RBD would impact the binding to one nanobody, but they don't provide any experimental support on the efficacy of the antibodies in recognizing and neutralizing the variants. They insist throughout the manuscript on the characteristics that make their nanobodies so adequate for clinical purposes, but they leave out proving that they will work against the current variants that are circulating in many countries. It is therefore important to provide at least some BLI data showing that the VHHs, at least, bind to the mutant RBDs (even better if they provide neutralization results, just binding to an isolated RBD may not recapitulate the action on the full Spike and neutralization of the infection).

We have now included large datasets that address this topic. Specifically, we re-selected our phage display libraries with a K417T E484K N501Y L452R quadruple RBD mutant to see which nanobodies tolerate all these mutations. Resistant ones (i.e., nanobodies that still bound to the mutant RBD) were then validated by BLI. Furthermore, we demonstrate low picomolar neutralization of the South African virus variant B.1.351. These data are shown in the new figures 9 and 10. Two of our initial leads (Re6H06 and Re9B09, which bind the main RBD epitope) as well as another Re9B09 class member turned out to be particularly potent against those mutant strains, even as monomers. In addition, we demonstrate very high anti-mutant potency of tandem fusions of an epitope 2-binder (Re9F06) with our Re5D06 lead or Re6H06. Likewise, the already characterized Re9F06 trimer turned out to neutralize B.1.351 with great potency too (actually more potently than the canonical strain).

- Regarding the analysis that they perform on the potential effects of the SARS-CoV-2 variants, there are some aspects that were left out of the discussion:

Lines 318-321: they don't expect changes in binding to the South African and Brazilian strains, because the aliphatic chain of the introduced K484 would make a favorable contact to Y109/W110 in the VHH. However, the E484 in the Wuhan strain forms hydrogen bonds with residues on the VHH and the authors don't discuss if these bonds can be formed with a lysine residue.

Thank you for making this interesting point. E484 is too far from any hydrogen bond-donor to form a hydrogen bond. The closest favorable interactions are to R50 (5.9-7.3Å) and R52 (4.7-6.6Å) of the nanobody Re5D06, with a localized water molecule between E484 and R52 (context shown below). The carboxy oxygens of E484 are actually closer to Y109 (3.4Å) and W110 (3.8Å), and we would assume that these interactions are favorable only because of the nearby (long-distance) ionic bonds.

The Rosetta software for protein structure predictions did not consider E484 to make a particularly strong contribution to affinity. Nevertheless, experimental testing now showed a clear reduction in affinity between Re5D06 and the South African or Brazilian mutant RBDs (new Fig 9). K_D s are now in the 1 nM range, which is still a pretty good affinity, but worse than we had expected from the initial modelling. The text has been revised accordingly. In addition, we now describe other nanobodies and nanobody-tandems that are highly resistant to the relevant mutations.

Moreover, they completely omit discussing the effect of the K417N/T mutation, when K417 makes a direct salt bridge with E113 in the nanobody (Fig. SI3). In sum, their discussion is incomplete and, possibly, too optimistic.

The RBD-K417/Re5D06-E113 salt bridge is rather peripheral and solvent-exposed, and the local electron density is weaker than expected for a stable interaction with full occupancy. When relaxing the structure in Rosetta, an alternative salt bridge of nanobody E113 to RBD R403 is preferred. We would therefore not be surprised if the K417 exchange had less of an impact than the mutation of the more central E484. Along this line, we added a note to the text to complete the discussion as requested.

- The only insights on the nanobodies neutralization mechanism come from the crystal structure they report. It is quite clear that the antibodies targeting epitope 1 will block ACE2 binding, but this is not so obvious for those directed towards epitope 2 (the overlapped region with ACE2 is rather small). The manuscript would benefit from competition experiments (which can be performed by BLI, a technique that the authors use) to experimentally demonstrate that blocking ACE2 binding is the mechanism of action of the panel of nanobodies, with particular emphasis on those against epitope 2.

We fully agree and have added the requested BLI data as the new Figure 5B. The outcome is that all tested epitope 1-binders block docking of the RBD to ACE2. Six of the epitope 2-binders (Re7E02, Re5F10, Re9F06/ Re6A11, Re6B07, Re6F06, and Re6H10) also compete, while another four (Re9C07,

Re9D02, Re9G12, and Re11H04) do not. This fits quite nicely our structure, which shows a frontal clash of the epitope 1-binder, while the epitope 2-binder Re9F06/Re6A11 produces a far smaller clash. A small shift in the footprint can then move the binder out of the clashing area.

- There is a very brief structural description on the epitope recognized by Re6A11. This seems a missed opportunity, since the authors report that the entire panel is recognizing basically two regions, and the focus of the analysis is put only on epitope 1. I think the authors should include a structural description of epitope 2 beyond the identification of contacting residues (Fig. 6d), as was done for Re5D06.

We have included a detailed display of the Re9F06-RBD interaction in Appendix Fig S5. The key interactions of Re9F06 with the RBD are, however, far less striking than those of Re5D06. This is in line with Re6A11/ Re9F06 binding 1000 times more weakly than the true high affinity binder Re5D06.

In this sense, the discussion would benefit from a comparison of both epitopes to those of previously reported nanobodies (is there any novel epitope, or everyone is isolating antibodies solely against two regions?) and mention if epitope 2 is conserved or not.

Epitope 2 is well conserved and so far, it is not hit by any of the common escape mutations (if we define this epitope as the footprint of Re6A11/ Re9F06). This is now extensively discussed. Except for an I to V and a peripheral P to A exchange, epitope 2 is even identical in SARS-CoV-1.

Looking at the other available structures, it seems that nanobodies of other groups target similar epitopes. VHH-72 (Wrapp et al. 2020) for example, which was initially selected against the SARS-CoV-1 RBD, recognizes this epitope. The same applies to nanobodies VHH-U, -V, -W of Koenig et al. (2021).

However, more of the available structures describe nanobodies targeting epitope 1. This applies, e.g., to NbH11-D4 and NbH11-H4 (Huo et al. 2020) as well as VHH-E (Koenig et al. 2021), which is the best nanobody of the latter study. One difference to our leads is, however, that VHH-E binds the RBD ~1000 times weaker and shows a 1000 times lower anti-viral potency than Re5D06. This is now discussed in the manuscript.

These considerations were added to the text. We also discuss a comparison of published anti-RBD nanobodies in terms of tolerance towards the common escape mutations (K417T/N, E484K, L452R). As judged from the available structures, it appears that the *best* leads of any previous study also target epitope 1 and are rather strongly impaired by these mutations.

- The structures should be deposited in the PDB (and the map in the EMDB) before submission and provide the corresponding codes.

Coordinates and structure factors for the Re5D06·RBD·Re9F06 complex and free Re5D06 have been deposited with the PDB (under IDs 7OLZ and 7ON5, respectively). We are in the process of submitting the three cryo-EM maps presented in Appendix Table S2 to the EMDB. The IDs will be added with the final edits of the manuscript text.

Similarly, the amino acid sequences of the 58 characterized nanobodies should be provided.

- The paragraph in lines 410-418 makes several statements for which the data are not provided. There should be a table listing the data on the different constructs mentioned and their IC₅₀, in particular to show the performance of the different collagen domains used and the corresponding linkers. Also, it is important to provide the amino acid sequences of the linkers used. Including all the data would better reflect the tremendous amount of protein engineering that was performed and would be very useful for the reader.

We agree and have added two tables (Appendix Tables S3 and S4) for selected nanobodies. As discussed above, we are providing IC₉₉ rather than IC₅₀ values, since we observed either complete or absent neutralization.

We are not aware of any related paper that has detailed all their sequences of nanobodies or monoclonal antibodies. So, this request goes quite far beyond the common practice. The sequences have been deposited in the patent database and will be released in ~6 months' time. We do not entirely object adding sequences to this manuscript, however, this will be a selection (of only fully characterized ones listed in Tables S3 and S4). As needed, we would be happy to discuss this topic further with the Editor, to find an appropriate balance between being transparent and considering that the current clinical development means a major financial investment that comes with constraints in terms of IP.

In addition, the authors should comment on the expected stability of the linker constructs vis-à-vis potential proteolysis (i.e., how stable are they with respect to fortuitous cleavage by circulating proteases when injected into a living organism)

We use G/S/E/D-rich linkers that are not particularly prone to proteolysis and lack typical tryptic/chymotryptic sites. Therefore, we do not expect much non-specific proteolysis in the plasma. The typical proteases of the coagulation cascade, complement activation, fibrinolysis, or collagenases recognize rather specific sequences that are not contained in our spacers. Furthermore, the plasma contains numerous potent protease inhibitors, such as α 1-antitrypsin or α 2-macroglobulin to suppress non-selective proteolysis.

Minor comments:

- The names used for the nanobodies are very hard to follow, sometimes they differ in a single number or a letter. I suggest that the authors come up with a better scheme. They could, for instance, use the domain and the epitope for those targeting the RBD (R1_01, R1_02, etc. and R2_01, R2_02, etc. for those targeting epitopes 1 and 2, respectively. And N01, N02, etc., for those targeting epitopes outside the RBD. The order (01, 02, could be an ascending order based on the measured K_d . Such a scheme would make the paper much easier to follow than it is currently the case.

We appreciate these suggestions, but for practical reasons, useful nanobodies need to have unique names. That is the only way to find them by search routines, be it for an intra-lab search or later for searching the Web.

Although we started with a large number, the reader does not need to memorize all of them. Only Re5D06, Re5F10, Re6B06, Re6H06, Re9B09, and Re9F06 occur repeatedly in the manuscript. We tried to simplify the text and to help the reader by supplying concise tables (Appendix Table S3 and S4). At places where we describe related nanobodies (e.g., the mutation-adapted Re9H03), we give explicit reference to the nanobody class. We simplified the nomenclature for the thermostable variants of Re5D06 to "R15" and "R28".

- The authors use the term "symmetry-matching" nanobodies (in the title and in several parts of the manuscript; lines 25, 76, 78, for example), but I am not sure this is accurate. The first time I read it I had the idea that it would refer to an object that matches exactly the position of the RBDs in the Spike (like a plug and a socket). However, the trimeric nanobodies present flexible linkers, so overall the molecule does not present a C3 symmetry. It would be more appropriate if the authors just called them "trimerized nanobodies", as they could also cross-link spikes at the virion surface, and bind in a non-symmetry matching pattern. If the authors wish to make a difference with respect to other strategies, for instance that VHHs linked via G4S linkers, they should state this difference and mention that their antibodies are trivalent and trimeric, while the ones produced with linkers are trivalent and monomeric.

We are happy to expand the term "trimeric" to "homotrimeric", whenever we mean three identical subunits. This should reduce ambiguity. We did not mean to imply that all subunits of a given spike are necessarily in the same conformation. However, we think that it is appropriate to consider their *architecture* as three-fold symmetric. We have tried to make the wording more clear.

- The authors propose their VHH+RBD co-expression strategy for a simplified production of vaccines and adaptation to viral escape-mutations (lines 29-30, 81-82, 263-277), which is debatable. First, if an

escape mutant appears the "folding" antibody may not bind anymore and that would be the end of the production system.

Thanks for challenging this perspective, but please allow us two answers to this point: (1) the fold promoter epitope has so far not been hit by any of the escape mutations, and we included this observation in our discussion. (2) We have several fold-promoting VHHs at hand (see Fig 5 and Appendix Table S3 for examples), and for each new mutant version, we would screen them for the best positive effect on RBD folding.

Second, the co-expression leads to co-purification, so the immunogen wouldn't be just the RBD, but the complex RBD+VHH.

This is accurate and intended. We are discussing in the manuscript that the co-purification of the VHH is not just a disadvantage:

" The presence of the fold-promoting nanobody as an RBD-ligand might provide additional benefits. Re9F06 leaves the "best epitope" for neutralization fully exposed; nevertheless, it prevents ACE2 from masking this epitope (Fig 6B) and should thereby improve its presentation to the immune system. This might also reduce the side effects of the immunization caused by an undesired binding of an RBD-vaccine to ACE2-presenting target cells with subsequent antibody-binding and thus opsonization of such cells."

In any case, we will make sure that the nanobody does not cover any neutralization-relevant mutant site.

- The term "force" is not well used (line 28, 79). It's better to use "promote", as was done in several sentences.

To avoid a mix-up with the physical term "force", we now use the verb "enforce".

- The wording in lines 54-58, which compare vaccines and passive immunization, is confusing, as it appears to suggest that the administration of antibodies could be done in a scale capable to stop the spread of a virus. This paragraph should be revisited.

Agreed. We meant spread within an infected person and not within a population. The paragraph has been amended as follows: *"Vaccination to raise antibodies against the Spike is the most widely used measure for blocking virus entry. It might, however, take two vaccinations and thus up to several weeks before a sufficient protective antibody level has built up. In contrast, passive immunization can take an immediate effect."*

- Line 66: "combinatorial issues" probably refer to combination of light and heavy chains, but it is ambiguous, as there are also 'combinatorial issues' at the VDJ recombination level.

To avoid such ambiguity, we get more specific about how the coding regions are cloned: *" This makes their coding regions straightforward to clone **from cDNA** (without combinatorial issues) into phage display vectors for subsequent selection of high-affinity binders"*.

- Was the affinity of anti-S1, non-RBD nanobodies determined? If so, they should be included in the supplementary Table requested above. This should also be mentioned to rule out that all these nanobodies are non-neutralizing only because they don't bind with high affinity.

We haven't measured their affinities. However, when staining spike structures in either transfected HeLa cells or infected Vero cells, we observed that the IF signal saturated at low nanomolar concentrations of these nanobodies. Likewise, the signal is not lost when the samples are washed for extended periods of time. We therefore assume that they are of reasonably good affinity.

We don't make a strong point out of the negative neutralization results, and do not claim that only the RBD provides neutralizing epitopes within the S1 fragment. We will probably revisit this topic in future experiments, but for the present paper we need to leave it at this.

Also, the terminology used to describe the methodology used to select them is confusing: S1 Δ RBD suggests that the authors have made a construct of S1 in which the RBD was deleted, which is not the case. Calling these Mabs just non-RBD binders is enough, as S1 was the only immunogen used to obtain the nanobodies.

We carefully considered these points but respectfully disagree. We need a concise term for the lettering of Figures, and "non-RBD binder" would not be a very intuitive assignment because this description can fit to any molecule except for RBD-binders and because the key information that these nanobodies recognize the S1 fragment would be lost. We would prefer to keep our current nomenclature.

Even though we obtained these binders through a subtractive selection (RBD-binders subtracted from the larger pool of S1 binders), the outcome is the same as if we had selected with an S1 Δ RBD domain. We have now put "S1 Δ RBD" initially in quotes to make it more clear that this is not a literal description but a definition of a technical term.

- Was the k_{on} and k_{off} measured by BLI or SPR? SPR is often a better method for obtaining kinetic parameters and it seems the authors used both, but the SPR was presented only on Fig. 7e.

Initial experiments were performed by SPR, but we then switched to BLI. One reason was throughput. The other (related) reason was that for several nanobodies SPR sensor chips could not be regenerated without damage. While this probably reflects their high affinities, it also made the SPR measurements prohibitively expensive as each datapoint meant that a new chip had to be used.

- Lines 176-183: unnecessary, it's quite obvious that the antibodies should neutralize at the lowest dose possible.

If the editor and reviewer agree, we would prefer to keep this paragraph as is. We feel that it does not simply make a trivial point but puts the issue of dose into a perspective (including that the Regeneron cocktail is administered in gram amounts). We think that this is useful information, especially for readers who are not familiar with the therapeutic doses used for, e.g., monoclonal antibodies.

- Lines 191-194: not necessary.

Again, we would prefer to keep this because this is the only place where we translate a neutralizing concentration into a therapeutic dose.

- Line 229: "trimeric complex". It is a ternary complex, not a trimer since the three objects in the complex are different.

Changed as suggested. "Ternary" is indeed more clear as it avoids confusion of the crystallized ternary complex with the homotrimeric nanobody fusions.

- The Materials and Methods section does not contain cloning information, and this should be included. Were the RBD, Re6A11 and Re5D06 cloned in three plasmids? In one plasmid? Which was the plasmid? Which was the plasmid used to express the other nanobodies?

The only crucial aspect was to co-express Re6A11/ Re9F06 with the RBD. This worked from two compatible plasmids, as a bi-cistron, or as a fusion. Re5D06 could either be co-expressed or added later. The actually used expression strategies are described in the text.

- Lines 249-262: I don't see a point on keeping this in the paper, but in case of doing so it should be on a separate Discussion section.

We removed the speculation that the nanobody promotes folding in a co-translation manner. That it suppresses association with GroEL and prevents a divergence to a non-productive folding pathway is, however, an observation. We therefore feel that it is as such well placed in the Results section.

- Lines 263-277: completely speculative and should be removed from the manuscript. The fact that co-expressing the VHH allowed the authors to obtain a complex and solve a structure is reason enough to include the strategy in the paper, there is no need to boost it with other debatable applications.

We gave this our careful consideration, and we appreciate that such statements may seem speculative. From our perspective, however, it is fair to state that there are more applications than just crystallizing a complex. A simple and economic way of producing a vaccine would be of high social impact. We would therefore rather leave this consideration in the manuscript, albeit with all due caution and clear emphasis on its hypothetical character.

- Lines 285-286: "cation- π interactions". According to Fig SI3 these interactions are between residues in the VHH, not with the RBD.

Thank you for pointing this out. Indeed, the figure missed F486 from the RBD. This has been amended. In fact, it is a cation- π relay, starting with R52/W110/R50 on the VHH side with R50 then making a third cation- π contact to F486 of the RBD.

- Lines 322-327 and Fig.6c: prepare the figure with the results of the experimental cryo-EM structure.

We have done so as outlined above. The previous panel 6C (modeling of docked Spike-Re5D06 complexes) has been moved to the Appendix (Fig S7).

- Line 356: it is not clear to me if Re5D06R13 has all the mutations mentioned in the previous paragraph, plus T58V and V93D. Does it have, in total, 10 mutations? Maybe including the list of all the changed residues in the figure could help, or as part of Materials and Methods (this applies to Re5D06R15 too).

We have clarified the text and the figure and present those details in Appendix Fig S12. The present version describes R15 and R28, as these turned out to be the most potent neutralizers amongst the hyperthermostable Re5D06 variants.

- Lines 372-374: the sentence regarding the potency should be toned down. The paper cited on reference 24 presents monomeric VHHs with IC₅₀ of 22 and 48pM (not the same as IC₉₉₊, but it sets a "precedent" among monomeric VHHs).

This has been re-phrased such that the combination of extreme anti-viral potency and hyperthermostability is unprecedented.

Xiang et al. do indeed describe very good nanobodies, and in particular their (related) leads Nb20 and Nb21 show very potent neutralization (as mentioned by the reviewer). However, both contact L452 and E484 in a manner that will leave them badly affected by the L452R and E484K mutations. The E484K mutation will probably completely eliminate binding. Following the suggestion of this reviewer, we also added this perspective to the text.

- Line 381: proposes a second disulfide on Re6H06 as the source of high thermal stability. Is this observed for all the antibodies with a second disulfide bond (Re5E03, Re5E11, Re5G05, Re6E11, Re6F06, etc.)? If not, remove this statement or discuss reasons for such a discrepancy.

So far, all tested nanobodies with a second disulfide bond showed the hyperthermostable phenotype (provided they were expressed in a way that ensures quantitative disulfide bond formation).

- Lines 385-388: probably match better the following paragraph.

We agree. These sentences have been moved down to the following section.

- Line 403-404: "it should be small for the sake of material economy". I don't understand this point... it should be small to save amino acids in the bacteria?? It seems better to delete this phrase.

This was meant as a serious point. At least in *E.coli*, the productivity goes down when using larger multimerization domains (productivity in terms of obtained protein mass per liter culture but also considering moles per liter culture).

- Lines 423-425: it is proposed that trimerization, instead of a dimeric Fc, is the responsible for a large increase in VHH-72 potency. Is this high-magnitude increase also obtained if VHH-72 is trimerized with collagen XV? If so, this favors the hypothesis, but if it's not the case then the statement should be revised.

We did not test the combination of VHH-72 with the collagen XV module (after having tested already >30 other trimers). To be more precise, we now specified the phrase to read: "Indeed, the collagen XVIII-trimerized VHH-72 neutralized down to 50 pM or 1.2 μ g/ liter".

- Lines 429-434: it seems an explanation on avidity, which every reader already knows.

We condensed this paragraph according to the suggestions, but we still feel that we need some explanation of why trimerization leads to a huge potency gain for low affinity nanobodies but to a much smaller one for nanobodies that bind with high affinity to begin with.

- Is the outstanding thermal stability maintained in the trimeric design?

In plain buffer, the collagen XVIII trimerization module is fully stable to $\sim 80^{\circ}\text{C}$. Given that collagens are extremely long-lived, we assume that this level of stability is sufficient.

- Overall, the "Outlook" section is too long, it should be changed into a Discussion and should be more focused.

We agree and have thoroughly re-written and shortened the section as suggested.

- Line 537: "66 kDa is well above... 60 kDa". It doesn't seem so "well above" to me. It's more like "barely the size...".

Agreed. It should have read "above" and not "well above". Taking serum albumin as a benchmark, a 60kDa protein can already have a quite respectable plasma half-life. Since we shortened this discussion, the size limit is no longer mentioned in the manuscript.

- Lines 544-547: "... it is improbable that a given mutant can escape many independent binders at the same time". I agree with this statement, but in the context of this article the different binders don't seem to be so "independent" since the 60 VHHS recognize basically two epitopes. Therefore, some single mutations could affect multiple binders at the same time. These considerations should be included in the discussion.

As outlined above, we addressed this issue experimentally – with strong support of our earlier statement. Note that the hypothesis is already confirmed if just a single neutralizing nanobody is identified that is not hit by a given escape mutation.

Epitopes 1 and 2 are non-overlapping and thus independent. We have looked at several nanobodies to epitope 1 and we found extreme differences in their sensitivity towards the current escape mutations. Re6D06, for example, is completely killed by either the South African or Californian mutations (shown in Appendix Fig S13). The affinity of Re5D06 is decreased to a ~ 1 nM KD, while Re6H06, Re9B09 or Re9H03 not only bind with low picomolar affinities (new Figure 9) but also neutralize the South African strain potently.

All epitope 2-binders tested are resistant to these mutations. For Re9F06, this is evident from our structure. High-affinity binding and potent neutralization data for Re5F10 are shown in Figure 9 and 10. Apart from answering this point, we wish to add that the "mutant section" has been completely revised.

- Line 562: among the examples of three-fold symmetry the authors mention Dengue and Zika viruses. They should be removed, since their envelope glycoproteins are displayed on the viral surface with different symmetry axes, they don't represent the trimeric arrangement seen on fusion proteins like those in Influenza, Coronaviruses, etc.

We deleted the reference to Dengue and Zika as suggested. It has been a side issue anyway.

- Fig.6 and Fig. SI3: it would be clearer if the dashed lines are used only to indicate polar contacts.

We understand, but prefer to keep the dashed lines for hydrophobic interactions as well. It mostly is a matter of personal preference. We like it because it gives you an impression of how extensive or dense a hydrophobic interaction network is (compare Appendix Fig S5 and S6).

- Figure 1: it would benefit of an inset in the figure with the color coding (it's easier than reading the figure legend).

We added this.

- Figure 8: include the theoretical molecular weight of each protein in their monomeric form (in this way, the reader can know that 60 kDa corresponds to a trimer).

We did this.

Typos:

- Line 113: extra parenthesis (or one missing).

- Line 115: "released from in the dissociation" (just "released" or "released from it").

- Line 280: ACE should be ACE2

All amended! And thankyou again for this deep, thorough, and very helpful review!

Referee #2

Güttler et al report the properties of nanobodies raised in alpaca by immunization with SARS-CoV-2 S1 fragment and SARS-CoV-2 RBD. They have found several that bind the RBD quite tightly (single-digit nM) and that neutralize (in a cell assay) with corresponding potency (ca 5-10 mg/L).

We respectfully disagree with this account of our data. We have found several nanobodies that bind (as monomers) the RBD with low picomolar affinity and that neutralize SARS-CoV-2 completely at concentrations below 1 $\mu\text{g}/\text{L}$. It is unclear to us why the reviewer quotes numbers that are 10 000 times worse.

This is the math: Lowest neutralizing concentration for the best monomer: 17-50 pM, molecular mass = 13 000 grams per mole. This brings us to 0.2-0.65 μg per liter. Best trimer: 0.6 pM, 66 000 gram per mole trimer. This brings us to 0.04 μg per liter. These numbers have been stated throughout the manuscript.

They also claim to have found even more tightly binding nanobodies, but see my concerns below. A structure of the RBD with two non-competing neutralizing nanobodies bound shows the tight fit and extensive solvent-excluding interfaces that give rise to high affinity. They have constructed a trimer of one of them, using a procollagen trimerization element, and as expected shown substantial increase in affinity and neutralization potency. Finally, because nanobodies can be expressed in good folded yield in E coli, but RBD ordinarily cannot, they find that at least one class of the nanobodies is an effective folding chaperone for the RBD.

The key problem with the paper is that it is all over the place, and flawed in each of those places -- probably not irreversibly, but enough to weaken substantially each part of it, leaving very little unflawed.

With all due respect, this blunt statement is not supported by the specific points that the reviewer has actually detailed. The reviewer correctly pointed out that the affinity data could be improved by measuring concentration-series of analytes. Such new data have now been added. We wish to note though that we arrived at rather similar numbers as before. Thus, our initial estimates have been valid. The point that our leads would neutralize only in the 10 mg per liter range is evidently incorrect (see above). Beyond this, this expert has not pointed out anything that could possibly be termed "flaw."

(1) The affinity measurements are flawed, because (unless Fig. 2 is misleading) they seem to have used only one concentration. The apparent dissociation rate constant is indeed low, but probably precisely because of mass transport -- which they seem to misunderstand. Mass transport will give apparently slower dissociation rate constants, because of vicinal rebinding while the ligand still contributes to the refractive index difference being detected. So unless I misread Fig. 2, the only ones of those numbers I can believe are those with nM KD and reasonably measurable koff, although I'm sure that the trimer does bind more tightly than the corresponding monomer.

We have included new data with BLI experiments being performed with concentration series of the analyte (new figures 2 and 9). The outcome is essentially the same, and the numbers did not change much. We still find low picomolar K_{DS} for our best monomeric RBD binders. While we agree that BLI cannot really discern absolute on- and off-rates, most systematic errors will cancel out when K_{DS} are computed. Determining precise K_{DS} in the low picomolar range is extremely challenging, but BLI (or SPR) are still the best of all methods with a reasonably high throughput.

We would also like to stress that our high affinity nanobodies fully resisted overnight off-rate selections with a more than 10 000-fold excess of free ligand over bait, which can be seen as a fully independent confirmation of their extremely high affinities.

(2) The structure analysis is fine as it stands, but only in the interest of further "drug design". We already know that there are several distinct immunogenic regions on the RBD. For example, the two Regeneron antibodies whose reputations were either enhanced or tarnished, depending on your point of view, by

being touted by infamous members of the former US administration, do not compete, and the crystal structure of those two antibodies bound together on the the RBD was reported almost a year ago. See also Piccoli et al, Cell, Nov. 20, 2020, and Barnes et al, Nature, Oct. 12, 2020, for very thorough analyses of sites of the RBD.

We are not sure why the analyses mentioned by the reviewer would compromise our structural analyses. This study did not aim at a general mapping of immunogenic regions on the RBD but simply outlined the preferred binding sites for the nanobodies generated in this particular experiment. We tried to remove any statement from our paper that raises the impression of too general statements.

(3) The authors make a reasonable claim for their preferred neutralization assay, but they do not seem to have validated it carefully. Was the nanobody removed before incubating the cells for 48 hrs -- otherwise, it could have been preventing spread, depending on the initial moi.

Indeed, we were trying to mimic the therapeutic situation and thus left the nanobody (and the inoculated virus) in the medium throughout the neutralization assays. We agree that we cannot exclude that nanobodies may interfere also with subsequent rounds of cell infections. However, we cannot see why this should be a shortcoming or even an indication for an inappropriate validation. We consider it more important that, at neutralizing concentrations, not a single cell expressed detectable amounts of virus proteins, and that such IF observations were fully consistent with orthogonal readouts such as RT-qPCR to quantify viral RNA or CPE observations (see Fig 4 and Fig S1 for examples).

(4) There are a number of statements that appear to show naiveté about immunology and antibody binding. (a) Divalent antibodies bind tightly not only because sometimes they can bridge two sites, but also because they rebind rapidly, presumably due to the presence of the second site at high local concentration. Most of us who work with both Fabs and IgGs find that the IgG from which the Fab was derived binds with 100 to 1000-fold lower K_D (more tightly), as measured by SPR or BLI, than does the Fab.

It is expected that a bivalent IgG shows a 100- to 1000-fold tighter binding in SPR or BLI than a monomeric Fab. We do not consider it naïve to explain this by avidity effects, which are nothing else than a re-binding while a second (third, fourth...) binding site is still engaged.

(b) There is no evidence for SARS-CoV-2 that most of the spikes need to be antibody bound to neutralize. For those viruses (e.g., influenza) for which detailed studies have been done, one needs far less than full occupancy, depending on the epitope.

The observation that full neutralization requires a nanobody concentration well (10 times) above the K_D is consistent with the assumption that full neutralization requires a majority of binding sites to be blocked. This is not just our observation. VHH-72, for example, binds the RBD with a K_D of 30 nM, while 500 nM are required for full neutralization. To be precise: we did not state that neutralization requires nanobody binding to all sites (=full occupancy) but just to a majority of sites. This is consistent with the data.

(c) The evolution of vertebrate immune systems (non-agnathan) is based on dimeric antibodies, so suggesting that dimers are not "good enough" seems a bit curious.

It was not just a suggestion but a combined observation that the VHH-72 trimer neutralized ~10 000 times better than a dimeric Fc fusion. Of course, dimers will be good enough if their affinity is high. For lower affinities, higher valencies are likely to help, which is in line with vertebrates initially producing IgAs and IgMs that have 4-12 binding sites (depending on the species).

Of course, from the point of view of designing a therapeutic, the trimer strategy is sensible (and has already been adopted and published for ACE2 as an inhibitor -- Xiao et al, NSMB Feb. 2021 but already in bioRxiv Sep 2020). (d) The required serum concentration of a therapeutic for reasonable clearance

will probably depend on serum lifetime, availability at the lung mucosa, etc., so the arguments about concentration, negligible elimination, etc., are oversimplified and not really relevant at this stage.

We agree that many parameters can influence the minimum effective serum concentration of a drug. Nonetheless, we would prefer leaving at least a rough discussion of the clinical perspectives in the discussion, and this includes at least the most basic parameters such as serum concentration and clearance.

(5) Production of the RBD in E coli because of the chaperone effect of the nanobody is the most interesting, perhaps the only really new, contribution. The rationale is a bit flawed -- RNA vaccines are almost instantly adaptable, as all one does is change the DNA template sequence, with little further production optimization needed, and the adapted vaccine can probably be tested expeditiously, as new phase 1 trials would be minimal or unnecessary because nothing would be different except the sequence of the RNA (and the expressed immunogen in the subject, so the issue of autoimmunity would always be there, but not generally detectable in a limited phase 1 trial anyway).

We agree with all these considerations. Still, it is highly unlikely that all vaccination problems can be solved by RNA vaccines in the near future. At least low-income countries with poor infrastructure will depend on more affordable alternatives, and a protein-based vaccine would fulfil this requirement, even when considering that higher regulatory hurdles need to be taken.

What this group should explore, given the finding and given who they are, is the general issue of whether co-expression with nanobodies can allow one to express single-domain eukaryotic proteins in E coli that cannot be obtained on their own in good yield. That would be a substantial advance and a lasting contribution.

Thank you for this comment. We do pursue the strategy of nanobody-assisted folding as a long-term commitment.

Moreover, the RBD alone, or even oligomerized on a scaffold, may not be an optimal immunogen. In the early, preclinical trials of the J&J vaccine, several different constructs were examined, including RBD, but intact spike was ultimately found to be preferable.

The RBD was in our hands exceedingly immunogenic, at least in alpacas. A good protein-based vaccine should indeed be multimeric. The context of a spike fulfills this boundary condition. However, there are technically simpler ways of producing multimers.

The J&J vaccine is not a protein-based but a vector-based vaccine that aims not only at humoral but also at cellular immunity. So, we do not consider this an appropriate comparison.

Referee #3

Guttler and colleagues have developed a series of single chain Llama antibodies raised against SARS-CoV-2 spike with a view to using them as therapeutics in COVID19. Fig 1 shows a phylogenetic tree of the Ab. Fig 2 shows binding data for the best Ab. Fig 3 and 4 show nM neutralisation data by immunofluorescent staining and QPCR to detect infected cells. Fig 5 uses IF neutralisation data to illustrate that the Ab bind 2 distinct areas on spike allowing 2 competition groups to be described. Fig 6 shows structures of the key Ab in complex with the spike RBD and shows how neutralisation is achieved. Fig 7 examines therostability and shows how this can be improved by mutagenesis of specific Ab. Fig 8 demonstrates how the Ab can be trimerized to improve neutralisation though improving affinity. The authors also show how co-expression of the Ab with Spike RBD improve RBD production and discuss how this may improve Antigen production for vaccines. The work is exciting, well presented and the experiments are compelling and well controlled.

We thank the referee for accurately summarizing our work and for the excellent overall evaluation.

I have only 1 major point, point 1, followed by minor suggestions to improve clarity.

1. There is no discussion of repeatability, how many times the experiments were repeated and how reliable this is. I'm sure its good but this needs to be shown. How representative are the neutralisation assays shown. These are usually plotted with error bars but here we just see one IF example. The QPCR bars are also apparently only done once with no error bars. Can we be reassured that this has been done more than once and some kind of statistical support be provided.

All neutralization experiments shown in the manuscript have been repeated at least once (in some cases more than 10-20 times), yielding the same results we present in the figures. We now include [n] for each dataset next to the fluorescence panels. Figures 3 and 4 not only describe the setup for our fluorescence-based neutralization readout, but they also serve to illustrate that the results from our imaging-based assay correlate well with the widely used (and more labor-intensive) method of detecting viral RNA in the cell culture supernatants by quantitative RT-PCR. For the examples shown, the culture supernatants of the imaged wells were processed and subjected to quantitative RT-PCR. Replicates would therefore only be technical replicates in these cases. (In fact, the average $c(t)$ value from two such technical replicates were used to determine the relative viral RNA load in the culture medium.) We agree with the reviewer that it is important to underscore that the fluorescence-based neutralization readout matches the respective quantitative RT-PCR data. We therefore show biological replicates (with both assays) for all datasets from Figures 3 and 4 in the Appendix (Figure S1). Note that in all cases the RNA load in SARS-CoV-2-positive wells is at least three orders of magnitude higher than that of SARS-CoV-2-negative wells.

2. The authors might want to discuss how their work contributes to future pandemic preparedness, ie having Ab against Coronaviruses ready to go therapeutically. Is there any comment they can make on broadness, ie their Ab hitting other coronas such as CoV1 or MERS?

The epitope 2 (recognized by Re9F06/ Re6A11) is so well conserved with SARS-CoV-1 (the only difference being a I to V and a peripheral P to A exchange) that we would expect cross-neutralization. MERS is more distant.

3. Line 44. I think that poor activity of remdesivir and direct acting antivirals is due to the fact that by the time someone shows up in hospital the virus has done the damage. Its not because remdesivir acts later in the lifecycle. I ask the authors to reconsider this point.

We agree, multiple factors contribute to the low efficacy of remdesivir against Covid-19. It has become clear that remdesivir is a less potent inhibitor of the SARS-CoV-2 RNA polymerase than initially hoped. And sure enough, it cannot directly revert damage caused by the virus or by the immune system. However, another point is that it cannot directly reduce an already accumulated virus load or prevent

viral entry into so far unaffected cells. We have revised the paragraph to avoid the impression of a single true explanation.

4. Para beginning line 131. This text is confusing because it talks about using fixed virus but then talks about infected cells, CPE and released virus. Please clarify how this was done.

We re-arranged the entire paragraph for more clarity. Essentially, we write how the infections were done and that we evaluate three different readouts for measuring infection/ neutralization (CPE, RT-qPCR, and IF for newly synthesized viral components). The newly established IF assay is now explained earlier in the manuscript.

5. The description of Fig 5 is hard to follow because there's no panels distinguished. Can you clarify this by giving the panels letters and referring to them individually. Generally what is competing with what could be more clearly described.

We have re-lettered the fluorescence images to improve clarity. Allocating panel letters (A-X), however, would not be practical in this case, because we show 32 competition experiments along with a non-competitor control (performed identically and in parallel). Thus, at least when sticking to the Latin alphabet, we would run out of letters.

Nevertheless, the figure now has a panel A and a panel B because we added BLI data for testing competition of nanobodies with ACE2 for RBD-binding.

6. From line 251 to 277 could be moved to discussion to focus on the data.

The paragraph has been condensed by removing the speculation of how the nanobody might promote RBD folding in a co-translational manner. We are now left with a paragraph describing observations.

Certainly more separation of results and discussion and some shortening of the discussion where possible would improve clarity. For example, more succinctly summarise the general advantages of nanobodies rather than review the field, and then focus on discussion of the specific Ab described herein.

Large parts of the text have been re-written according to the reviewers' suggestions and in particular to accommodate the new large dataset on nanobodies that are resistant to the viral escape mutations. Thus, large parts of previous discussion elements are now replaced by data.

7. Outlook should be discussion I guess.

We have condensed and re-written the previous Outlook section and merged it with the Results & Discussion section. Having a separate Discussion, revisiting even just the most important findings, is not going to work because the manuscript is already extremely long (105 000 characters without Supplements)

Typos/grammer

8. Line 34 the virus infected more (no had)

9. Line 55 However, to be effective....

10. Line 115 released in the (no from)

11. Line 538 cut off at

These typos have been corrected, thank you!

Dear Dirk,

Thank you for submitting your revised manuscript. Your study has now been re-reviewed by referees #1 and 3. As you can see from the comments below both referees appreciate the introduced changes and support publication here.

I am therefore very happy to accept the manuscript for publication here. There are just a few formatting issues to resolve before I can send you the formal acceptance letter.

- Our publisher has done their pre-publication checks on the MS. Their comments are visible in the word document called "Data edited MS file". Please respond to their comments regarding figure legends and data availability section.
- Please double check that you have uploaded high resolution figure files
- We are also missing 3-5 keywords.
- The author contribution is also missing
- For the reference list: for citations with more than 10 authors please reduce to 10 et al.
- I think a figure callout is missing for Fig 8B
- Some of the blots show in the appendix like for example Fig S4 are very contrasted and low resolution. I think it would make sense to provide the source data for the blots shown in the appendix as separate source datafiles so that the reader will have access to the "raw data". Please upload the source data as one file per figure.
- We include a synopsis of the paper on the html version (see <http://emboj.embopress.org/>). Please provide me with a general summary statement and 3-5 bullet points that capture the key findings of the paper.
- We also need a summary figure for the synopsis. The size should be 550 wide by [200-400] high (pixels). You can also use something from the figures if that is easier.

That should be all. Let me know if you have any further questions.

Thank you

Karin

Karin Dumstrei, PhD
Senior Editor
The EMBO Journal

The revision must be submitted online within 90 days; please click on the link below to submit the

revision online before 15th Sep 2021.

Referee #1:

The revised version addresses all of the issues I have raised. The paper is still quite long, but with the amount of data that they describe, it is unavoidable. The revised version is indeed much stronger than the initial one, especially now that they have evaluated the identified nanobodies against the new variants of concern of the SARS-CoV-2.

Referee #3:

The authors have effectively addressed my comments, and those of the other reviewers, and in my view the manuscript is now a very nice piece of work in an important area and is compelling, effectively performed and controlled, and nicely written. I have no further suggestions.

Dear Dirk,

Thank you for submitting your revised manuscript to The EMBO Journal. I have now had a chance to take a look at everything and all looks good.

I am therefore very pleased to accept the manuscript for publication here.

Congratulations on a great paper!

best Karin

Karin Dumstrei, PhD
Senior Editor
The EMBO Journal

Please note that it is EMBO Journal policy for the transcript of the editorial process (containing referee reports and your response letter) to be published as an online supplement to each paper. If you do NOT want this, you will need to inform the Editorial Office via email immediately. More information is available here:

<https://www.embopress.org/page/journal/14602075/authorguide#transparentprocess>

Your manuscript will be processed for publication in the journal by EMBO Press. Manuscripts in the PDF and electronic editions of The EMBO Journal will be copy edited, and you will be provided with page proofs prior to publication. Please note that supplementary information is not included in the proofs.

Please note that you will be contacted by Wiley Author Services to complete licensing and payment information. The 'Page Charges Authorization Form' is available here:

https://www.embopress.org/pb-assets/embo-site/tej_apc.pdf

Should you be planning a Press Release on your article, please get in contact with embojournal@wiley.com as early as possible, in order to coordinate publication and release dates.

If you have any questions, please do not hesitate to call or email the Editorial Office. Thank you for your contribution to The EMBO Journal.

Corresponding Author Name: Dirk Goerlich

Journal Submitted to: The EMBO Journal

Manuscript Number: EMBOJ-2021-107985R